
# Statistical Characterization of Environmental Hot Spots and Hot Moments and Applications in Groundwater Hydrology

Jiancong Chen[1], Bhavna Arora[2], Alberto Bellin[3] and Yoram Rubin[1]

[1]Department of Civil and Environmental Engineering, University of California, Berkeley, California, USA
[2]Energy Geosciences Division, Lawrence Berkeley National Laboratory, Berkeley, California, USA
[3]Department of Civil, Environmental and Mechanical Engineering, University of Trento, Italy

*Correspondence to*: Yoram Rubin (rubin@ce.berkeley.edu)

**Abstract**

Environmental hot spots and hot moments (HSHMs) represent rare locations and events that exert disproportionate influence over the environment. While several mechanistic models have been used to characterize HSHMs behavior at specific sites, a critical missing component of research on HSHMs has been the development of clear, conventional statistical models. In this paper, we introduced a novel stochastic framework for analyzing HSHMs and the uncertainties. This framework can easily incorporate heterogeneous features in the spatiotemporal domain and can offer inexpensive solutions for testing future scenarios. The proposed approach utilizes indicator random variables (RVs) to construct a statistical model for HSHMs. The HSHMs indicator RVs are comprised of spatial and temporal components, which can be used to represent the unique characteristics of HSHMs. We identified three categories of HSHMs and demonstrated how our statistical framework are adjusted for each category. The three categories are (1) HSHMs defined only by spatial (static) components, (2) HSHMs defined by both spatial and temporal (dynamic) components, and (3) HSHMs defined by multiple dynamic components. The representation of an HSHM through its spatial and temporal components allows researchers to relate the HSHM's uncertainty to the uncertainty of its components. We illustrated the proposed statistical framework through several HSHM case studies covering a variety of surface, subsurface, and coupled systems.

## 1 Introduction

Environmental hot spots and hot moments (HSHMs) were originally defined as rare locations or events that support or induce disproportionately high activity levels (e.g., chemical reaction rates) compared to surrounding areas or preceding times (McClain et al., 2003). Vidon et al. (2010) further classified HSHMs into either transport-driven or biogeochemically-driven HSHMs, based on the mechanisms causing the HSHMs. Bernhardt et al. (2017) derived the concept of ecological control points (CPs) related to HSHMs, defining CPs as areas of the landscape that exert a disproportionate influence on the biogeochemical behavior of an ecosystem under study. These definitions have mainly focused on HSHMs related to elevated biogeochemical activities triggered by hydrological or biogeochemical processes, or a confluence of both processes. The concept of HSHMs is also used in climate science, where it is related to elevated greenhouse gas emissions or specific locations that are subject to extreme natural hazards (e.g., sea-level rise, floods, hurricanes, or earthquakes) caused by climate change (Arora et al., 2020; Shrestha and Wang, 2018). Further, Henri et al. (2015) related HSHMs to locations experiencing elevated environmental risks and developed the





incremental lifetime cancer risk (ILCR) model to quantify the effects of hot spots on human health. Overall, these studies have focused on quantifying the consequences of HSHMs by way of environmental risks and costs while also emphasizing the importance of characterizing the occurrences of environmental HSHMs. In the present study, we

combined these definitions such that, henceforth, HSHMs are referred to as rare locations or events that could exert a disproportionate influence on an ecosystem and which are associated with heightened health or environmental risks.

Characterizing HSHMs dynamics is useful for understanding hydrological and ecological dynamics related to nutrient cycling, contaminant transport, and accurate assessment of ecosystem and hydrological perturbations under climate change. For example, Duncan et al. (2013) demonstrated that riparian hollows, which represent less than 1.0%

of the landscape but contribute to more than 99% of total denitrification of a whole catchment area, function as hot spots. Additionally, wetlands have been considered biogeochemical hot spots for mercury mobilization and methylation production since the early 1990s (Vidon et al., 2010). The spatial patterns of methylmercury (MeHg) hot spots in wetlands can vary significantly across space. Indeed, the MeHg concentration at the interface between upland and peatland can be 100 times greater than a different patch within the same wetland (Mitchell et al., 2008). In

managed temperate peatlands, drainage ditches that account for less than 5% of a land area can act as hot spots and can contribute to over 84% of total greenhouse gas emissions (Teh et al., 2011). The disproportionate contributions from HSHMs to the overall hydrological and ecological dynamics strongly indicate the necessities of characterizing HSHMs.

Quantifying HSHMs has also been recognized as important for assessing the consequences after catastrophes

and the environmental risks, such as water crises (Baum et al., 2016) or nuclear disasters  (Kamidaira et al., 2018; Morino et al., 2011; Showstack, 2014). The migration of contaminants after a catastrophe creates zones of different toxicity levels and poses disproportionate threats to the surrounding natural and urban environment. In contrast, existing HSHMs caused by the leakage of nuclear waste or heavy metals largely influence site characterization needs and the remediation efforts needed to minimize environmental and economic losses (Bao et al., 2014; Harken et al.,

2019). Thus far, studies in this area have focused on the environmental implications and usefulness of characterizing HSHMs. However, special tools for characterizing and modeling HSHMs are still needed, such as physically-based and statistical models, which can provide additional benefits to capture the disproportionate effects of an HSHM on a whole ecosystem.

Reactive transport models have been used to understand and predict HSHMs. Dwivedi et al. (2017), for

instance, developed a 3-D high-resolution numerical model to investigate whether organic-carbon-rich and chemically-reduced sediments located within the riparian zone act as denitrification hot spots. Their study demonstrated a significantly higher potential (~70%) of the naturally reduced zones (NRZs) to remove nitrate than the non-NRZ locations. Arora et al. (2016) used a 2-D transect model and showed that temperature fluctuations constituted carbon hot moments in a contaminated floodplain aquifer that resulted in a 170% increase in annual groundwater

carbon fluxes. Gu et al. (2012) developed a Monte Carlo reactive transport approach and discovered how denitrification HSHMs are triggered by river stage fluctuations. Despite these studies, clear statistical conventions of HSHMs are missing, which significantly limits the transferability of these approaches. In fact, distinguishing HSHMs



based on statistical formulations has been identified as a major gap in the current HSHM literature (Bernhardt et al., 2017; Arora et al., 2020).

Statistical approaches offer multiple advantages for furthering the HSHM concept. First, statistical approaches can develop common formulations that integrate biogeochemical and hydrogeological knowledge from multiple HSHMs studies. Once developed, these formulations can be readily applied to identify HSHMs at similar sites. Second, statistical approaches can easily incorporate categorical indicators that represent spatial heterogeneity and quantify the uncertainty of HSHM occurrences tied to these features. Such approaches can be used as predictive

tools to estimate future occurrences of HSHMs, and provide an alternative to computationally-expensive high-resolution mechanistic models. This would greatly aid decision-makers in identifying scenarios (e.g., changes in the climate or in environmental conditions) that increase risks associated with the occurrence of HSHMs phenomena.

        Statistical concepts and models have been widely applied in hydrology and hydrogeology, including but not limited to modeling flow and contaminant transport, quantifying subsurface heterogeneity and the associated

uncertainties, developing strategies for site characterization, and providing informative priors for ungauged watersheds. For example, Rubin (1991) described a Lagrangian approach to obtain the spatial and temporal moments of contaminant concentrations in the subsurface. These statistical moments were deemed both necessary and sufficient to define the probability distribution of contaminant concentrations over space and time, and thus, quite useful for quantifying HSHMs. In a similar manner, statistical moments can be used to characterize the occurrences of HSHMs.

Statistical terms, such as concentration mean and variance, concentration cumulative density function (CDF), exceedance probabilities, and exposure time CDF also provide significant guidance to assess the environmental risks associated with HSHMs (Rubin et al., 1994). Although there is a lack of conventional statistical approaches in current HSHM studies, we believe it is feasible and valuable to develop statistical formulations to characterize HSHMs dynamics.

Successful characterization of HSHMs through physically-based models or statistical approaches relies on experts' knowledge of a site, intensive field characterization, and possibly continuous field sampling to provide the data to develop and validate these approaches. Understandably, intensive site characterization and long-term sampling can be quite challenging due to the associated costs and efforts. Thus, it is necessary to develop approaches that could simplify but still effectively and efficiently represent the underlying structure of HSHMs. In this regard, indicator

statistics, defined by the Bernoulli distribution, can be useful, on two counts. First, it is suitable for modeling bimodal situations. For example, a situation where an event might or might not take place. Indicators are also appealing in applications because of the sparsity of the Bernoulli probability model. Indicator statistics have previously been applied to model flow and transport phenomena in groundwater (Rubin and Journel, 1991), where indicators were used to model the spatial distribution in a sand-shale formation. Rubin (1995) applied an indicator spatial random

function to model contaminant flow and transport in bimodal heterogeneous formations. Ritzi et al. (2004) developed a hierarchical architecture to represent the spatial correlation of permeability in cross-stratified sediment using indicator statistics. Wilson and Rubin (2002) and Bellin and Rubin (2004) used indicator statistics that describe whether particles were captured by sampling points to characterize the level of aquifer heterogeneity. These studies suggest that the simplification of the system's structure through indicator formulation significantly lower the number



of measurements needed, and thus reduce the costs associated with site characterization, while maintain sufficient information for modeling flow and contaminant transport. In addition, indicator formulation is useful in that it allows to aggregate multiple variables (e.g., all HSHM relevant variables) into a single random variable. Instead of characterizing the full distributions of each parameter, indicator formulation only requires knowledge of the critical condition for relevant parameters. Such indicator RV will take a value of 1 if the critical conditions are met, regardless

of the original distribution for the parameters. These advantages are further explored in section 2 and 3. With indicator formulations for HSHMs, researchers can focus on identifying the most relevant parameters for HSHMs quantification, which can significantly reduce the efforts and costs required for intensive site characterization.

In this study, we developed a statistical framework to quantify HSHMs occurrences and uncertainties. The developed statistical framework can help determine HSHM-occurrence probabilities under user-defined scenarios. It

can also be used for estimating future occurrences of HSHMs. Based on the mechanisms that drive HSHM occurrences, we determined three categories of HSHMs: (1) those triggered only by spatial (static) contributors, (2) those triggered by both spatial (static) and temporal (dynamic) contributors, and (3) those triggered by multiple dynamic contributors. Within each category, cases from existing studies were used to illustrate the procedures for constructing the statistical formulations. We focused specifically on HSHMs applications in groundwater, where we

derived analytical solutions for the statistical formulation of HSHMs and analyzed the probabilities of HSHM occurrences and their corresponding levels of uncertainty using synthetic case studies.

## 2 Statistical formulation of hot spots and hot moments

HSHMs represent rare places or events with increased hydrobiogeochemical rates or fluxes that are significantly elevated above the background condition, thus exerting disproportionate influences over an ecosystem's

dynamics. We define $(\boldsymbol{\Omega}^*, t^*)$ as the jointly distributed RVs for HSHMs, and $\boldsymbol{\Omega}^*$ and $t^*$ represent the spatial components of hot spots and temporal components of hot moments, respectively. An indicator random variable, $I_{HSHM}(\boldsymbol{\Omega}^*, t^*)$, is used to represent whether the pair $(\boldsymbol{\Omega}^*, t^*)$ is an HSHM or not. If there exists a pair of $(\boldsymbol{\Omega}^*, t^*)$ that satisfies the critical conditions of an HSHM, $I_{HSHM}(\boldsymbol{\Omega}^*, t^*) = 1$, and the pair $(\boldsymbol{\Omega}^*, t^*)$ represents the location and time of the HSHM.

Following the original definition by McClain et al. (2003), in our method, $I_{HSHM}(\boldsymbol{\Omega}^*, t^*)$ can take the value of 0 or 1, depending on the concentration or reaction rate measure at $(\boldsymbol{\Omega}^*, t^*)$, respectively:

$$I_{HSHM}(\boldsymbol{\Omega}^*, t^*) = \begin{cases} 1, if \ C(\boldsymbol{x}, t^*) > C_{th}; \ \boldsymbol{x} \subseteq \boldsymbol{\Omega}^* \\ 0, \qquad otherwise \end{cases}, \text{or}$$

$$I_{HSHM}(\boldsymbol{\Omega}^*, t^*) = \begin{cases} 1, if \ R(\boldsymbol{x}, t^*) > R_{th}; \ \boldsymbol{x} \subseteq \boldsymbol{\Omega}^* \\ 0, \qquad otherwise \end{cases}. \quad (1)$$

where $C(\boldsymbol{x}, t^*) \ and \ R(\boldsymbol{x}, t^*)$ are the concentration and reaction rate at the position $\boldsymbol{x}$ and time $t^*$, respectively.

$C_{th} \ and \ R_{th}$ represent the concentration and reaction rate thresholds, respectively. Defining indicators with concentration, or reaction rate depends on the target of HSHM. Similar definitions can be introduced based on the regulatory limits or the interest of the investigator, using the mean concentration or the solute mass within the volume $\boldsymbol{\Omega}^*$.





The critical values, $C_{th}$ and $R_{th}$, are key to an effective application of the above framework and should be

determined based on the specific scenario. For example, for contaminants that are associated with significant environmental or health risks (e.g., nuclear waste or a cancerous substance), $C_{th} = 0$ can be used so that the HSHM will be triggered as soon as there is the presence of such contaminants. As an alternative, a limit in the total accumulated mass within hot spots may be set, such as suggested by EPA (USEPA, 2001), but in this case the definition (1) of the indicators should be modified. For water quality parameters, $C_{th} = MCL$ can be assigned, where

$MCL$ represents the maximum concentration limit for a specific solute. Alternatively, $C_{th} = C^*$ can be used in cases where $C^*$ is chosen based on the experts' domain knowledge. This approach requires that such decisions be made before deriving any solutions to determine HSHM occurrences.

Given the definition of $I_{HSHM}(\boldsymbol{\Omega}^*, t^*)$, we observe that $I_{HSHM}(\boldsymbol{\Omega}^*, t^*)$ follows a Bernoulli distribution, such as $I_{HSHM}(\boldsymbol{\Omega}^*, t^*) \sim Bernoulli(< I_{HSHM}(\boldsymbol{\Omega}^*, t^*) >)$, where $< . >$ is the operator indicating the ensemble mean of the

indicator represented as a random variable. An important characteristic of the Bernoulli distribution is that all the statistical moments of the RV $I_{HSHM}(\boldsymbol{\Omega}^*, t^*)$ can be expressed as a function of the ensemble mean $< I_{HSHM}(\boldsymbol{\Omega}^*, t^*) >$. For example, the variance is given by $var(I_{HSHM}(\boldsymbol{\Omega}^*, t^*)) = < I_{HSHM}(\boldsymbol{\Omega}^*, t^*) > \cdot (1 - < I_{HSHM}(\boldsymbol{\Omega}^*, t^*) >)$.

Characterization of the spatiotemporal distribution of $I_{HSHM}(\boldsymbol{\Omega}^*, t^*)$ requires the incorporation of the mechanisms that govern the development and occurrence of HSHMs. However, the direct quantification of $<$

$I_{HSHM}(\boldsymbol{\Omega}^*, t^*) >$ can be difficult in both time and space domain. Thus, to facilitate this undertaking, we propose to decompose $I_{HSHM}(\boldsymbol{\Omega}^*, t^*)$ into a Type-A (static) indicator random variable—$I_s(\boldsymbol{\Omega}^*)$—and a Type-B (dynamic) indicator random variable—$I_d(\boldsymbol{\Omega}^*, t^*)$. Definitions of the Type-A and Type-B contributors are as follows:

- **Type-A (Static) Contributors.** This category covers discrete spatial elements (and their associated critical states) that could trigger an HSHM once they come into contact with Type-B contributors (see discussion
below). Critical states are the range of values needed to trigger an HSHM (either in standalone mode or when coupled with Type-B contributors).

- **Type-B (Dynamic) Contributors**. This category covers dynamic variables (and their associated critical states) that could trigger an HSHM once they come into contact with Type-A contributors. This category includes, for example, mass transport variables. It also includes changes in local hydrological and
environmental conditions (e.g., water table fluctuations). The displacement of solutes in the subsurface (trajectories and travel times) from below- and above-ground processes are prime examples of Type-B contributors.

As an example, naturally reduced sediments (Type-A contributor) occurring next to the river corridor at the Rifle site were identified as carbon export hot spots (Arora et al., 2016; Wainwright et al., 2015). Studies showed that

these hot spots were triggered when temperature conditions (Type-B contributor) varied in the subsurface, resulting in a 170% increase in groundwater carbon export from the floodplain site to the river (Arora et al., 2016). In another example, topographic features, such as the backslope of the lower montane hillslope (Type-A contributor) within the East River Watershed (Hubbard et al., 2018), were considered denitrification hot spots, which can have a significant





impact on the watershed-scale nitrogen loss pathway. These hot spots were often triggered by spring snowmelt and
storm events (Type-B contributor).

Both indicators of the Type-A and Type-B contributors assume a value of either 0 or 1. If one of these
indicators takes a value of 1, it can be viewed as an HSHM contributor. However, for an HSHM to occur, both
indicators must have a value of 1 at the same location and time. This idea can be expressed as follows:

$$P(I_{HSHM}(\boldsymbol{\Omega}^*, t^*) = 1) = P(I_s(\boldsymbol{\Omega}^*) = 1, I_d(\boldsymbol{\Omega}^*, t^*) = 1)$$

$$= P(I_s(\boldsymbol{\Omega}^*) = 1) \cdot P(I_d(\boldsymbol{\Omega}^*, t^*) = 1 | I_s(\boldsymbol{\Omega}^*) = 1)$$

$$= P(I_d(\boldsymbol{\Omega}^*, t^*) = 1) \cdot P(I_s(\boldsymbol{\Omega}^*) = 1 | I_d(\boldsymbol{\Omega}^*, t^*) = 1). \quad (2)$$

In Eq. (2), $P(I_d(\boldsymbol{\Omega}^*, t^*) = 1 | I_s(\boldsymbol{\Omega}^*) = 1)$ is the probability of observing a dynamic HSHM within $\boldsymbol{\Omega}^*$, at
time $t^*$ conditional to the fact that $\boldsymbol{\Omega}^*$ is a static hotspot and $P(I_s(\boldsymbol{\Omega}^*) = 1 | I_d(\boldsymbol{\Omega}^*, t^*) = 1)$ is defined similarly. Based
on the mechanisms of HSHMs, we can classify HSHMs into three different categories as discussed below. These
categories can be used to guide the application of the above statistical framework in a variety of complex HSHM
scenarios, and they can also be used to develop analytical or numerical solutions for both static and dynamic indicators.
Furthermore, the three categories provide guidance on using indicator approaches for both transport-driven and
biogeochemically-driven HSHMs, as discussed by Vidon et al. (2010).

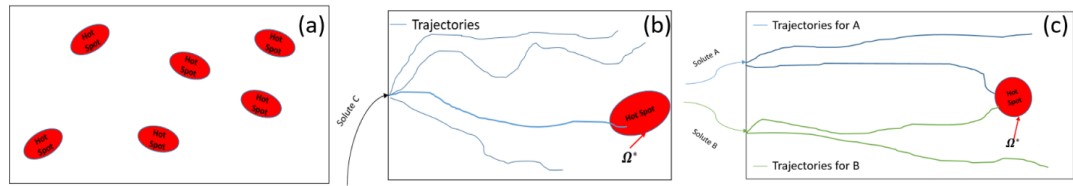

Figure 1. Identified categories of HSHMs. Panel (a) presents HSHMs resulting from only Type-A (static) indicator;
panel (b) presents HSHMs resulting from coupled action (static + dynamic) and panel (c) presents HSHMs resulting
from multiple (two) dynamic indicators

### 2.1 HSHMs induced by type-A (static) indicators

In this section, we consider HSHMs that are defined by static indicators only (Figure 1a). This list can include
zones of high, persistent concentration and reactivity that are due to the subsurface or the ecosystem's unique
hydrological and biogeochemical properties. For example, the accumulation of contaminants in the subsurface (e.g.,
the high nuclide concentration in the subsurface at the Hanford site) could lead to the evolution of persistent, high
reactivity zones. An aquifer's reactivity is another example that could distinguish certain regions with high reactivity
compared to surrounding areas (Loschko et al., 2016). Such high reactivity spots (hereafter denoted as $\boldsymbol{\Omega}^*$) can be
characterized by static indicator RVs due to the persistence of high concentration or reactivity. The static indicators
are defined as follows:

$$I_{HSHM}(\boldsymbol{\Omega}^*) = I_s(\boldsymbol{\Omega}^*) = \begin{cases} 1, & if \ Z(\boldsymbol{\Omega}^*) \subseteq Z_s^* \\ 0, & otherwise \end{cases}, \quad (3)$$

where $Z_s^*$ represents the conditions needed to trigger a hot spot at $\boldsymbol{\Omega}^*$, and $Z(\boldsymbol{\Omega}^*)$ represents the corresponding local
conditions at $\boldsymbol{\Omega}^*$.





**2.2 HSHMs induced by type-A (static) and type-B (dynamic) indicators**

HSHMs can also result from dynamic processes encountering specific conditions at $\boldsymbol{\Omega}^*$ (Figure 1b). This is the situation described by Eq. (2), where the static indicators are determined first, and then used jointly with the dynamic indicators for complete HSHM characterization. For example, Bundt et al. (2001) concluded that preferential flow paths are biological hot spots for soil microbial activities. Preferential flow paths in such cases are candidate hot

spot locations ($\boldsymbol{\Omega}^*$). Meanwhile, dynamic factors, such as snowmelt, control contaminant transport via the preferential flow paths, and thus, they determined the hot moment component. The duration of these events presents the temporal component of the HSHM.

For an HSHM induced by both static and dynamic indicators, the static locations are selected first, based on their HSHM-related properties. After this, we can focus on characterizing the HSHM dynamics as they relate to the relevant

locations. A selected location, $\boldsymbol{\Omega}^*$, could become an HSHM site based on characteristics defined through the following static and dynamic indicators, respectively:

$$I_s(\boldsymbol{\Omega}^*) = \begin{cases} 1, & if\ Z_s(\boldsymbol{\Omega}^*) \subseteq Z_s^*, \\ 0, & otherwise \end{cases}, \qquad (4)$$

$$I_d(\boldsymbol{\Omega}^*, t^*) = \begin{cases} 1, & if\ Z_d(\boldsymbol{\Omega}^*, t^*) \subseteq Z_d^*, \\ 0, & otherwise \end{cases}, \qquad (5)$$

$$I_{HSHM}(\boldsymbol{\Omega}^*, t^*) = \begin{cases} 1, & if\ Z_s(\boldsymbol{\Omega}^*) \subseteq Z_s^*, and\ Z_d(\boldsymbol{\Omega}^*, t^*) \subseteq Z_d^*, \\ 0, & otherwise \end{cases}, \qquad (6)$$

where $Z_d^*$ represents the critical conditions needed to characterize a hot moment, and $Z_d(\boldsymbol{\Omega}^*, t^*)$ represents the local condition at $t^*$ and $\boldsymbol{\Omega}^*$. The statistical model of $I_{HSHM}(\boldsymbol{\Omega}^*, t^*)$ can be expressed using the statistical models of $I_s$ and $I_d$, as shown in Eq. (2).

**2.3 HSHMs induced by multiple type-B (dynamic )indicators**

Various dynamic processes could jointly evolve into an HSHM (Figure 1c). Unlike the previous scenarios

where static locations can be determined through known characteristics provided by geophysical or other types of data, HSHMs can also emerge due to the confluence of dynamic processes. This situation is described in Eq. (7). For example, Gu et al. (2012) analyzed how streamflow fluctuations could trigger a nitrogen HSHM. In their example, the dynamics of the streamflow and groundwater controlled the transport and mixing of the chemical reactants, thus triggering the occurrences of HSHMs. For this case, the static locations of $\boldsymbol{\Omega}^*$ are determined by the confluence of

multiple dynamic processes.

We can consider the case where an HSHM is predicated on $m$ dynamic processes, $d_j$, where $I_{d,j}(\boldsymbol{\Omega}^*, t^*)$ is the dynamic indicator representing the action (or inaction) of $d_j$ at $\boldsymbol{\Omega}^*$ and time $t^*$. The hot spot location $\boldsymbol{\Omega}^*$ is determined by the confluence of all dynamic processes at time $t^*$. These dynamic processes are not necessarily independent. Therefore, generally, the statistical model for the comprehensive dynamic indicator (which covers all

dynamic contributors) assumes the following form:

$$P[I_d(\boldsymbol{\Omega}^*, t^*) = 1] = P[I_{d,1}(\boldsymbol{\Omega}^*, t^*) = 1, \dots, I_{d,m}(\boldsymbol{\Omega}^*, t^*) = 1]. \qquad (7)$$

In situations where the various dynamic contributors can be viewed as independent (e.g., Destouni and Cvetkovic, 1991)—where the reactants travel via different paths—then, assuming independence, we can state that



$$P(I_d(\boldsymbol{\Omega}^*, t^*) = 1] = \prod_{j=1}^{m} P\big[I_{d,j}(\boldsymbol{\Omega}^*, t^*) = 1\big]. \tag{8}$$

Here, the mean of the dynamic indicator becomes

$$< I_d(\boldsymbol{\Omega}^*, t^*) > = \prod_{j=1}^{m} < I_{d,j}(\boldsymbol{\Omega}^*, t^*) >. \tag{9}$$

If $\boldsymbol{\Omega}^*$ is a hot spot, then Eq. (9) also defines $< I_{HSHM}(\boldsymbol{\Omega}^*, t^*) >$. However, if $\boldsymbol{\Omega}^*$ is not a hot spot, then we need to resort to coupled statistical modeling, as suggested by Eq. (2).

**3 Examples of the statistical formulation of HSHMs with case studies**

250        In this section, we selected numerous examples from published research to present how our approach can be used to derive statistical representations for the HSHMs investigated in these studies. We grouped these studies into three categories based on the similarities of their underlying HSHM mechanisms, as described in section 2. We also characterized the environmental risk levels and impacts based on their target HSHMs. Table 1 presents a summary of these cases. The indicator formulation is constructed in sections 3.1–3.3.



| Reference | HS Location | Category | Seasonality | Environmental Risk | Causes | Impact | Static Mechanism | Dynamic Mechanism | HSHM Action | Metrics for threshold | Equation(s) |
|---|---|---|---|---|---|---|---|---|---|---|---|
| **Examples of static only mechanisms** | | | | | | | | | | | |
| Wainwright et al. (2015) | Naturally reducing zone | Subsurface | -- | Short-term low risk; long-term high risk | Anthropogenic + Natural | Negative | Mineralological and lithological differences | -- | Vanadium, uranium, metallic minerals | Concentration | (3) |
| Sassen et al. (2012) | Reactive facies | Subsurface | -- | Short-term low risk; long-term high risk | Anthropogenic + Natural | Negative | Lithological differences | -- | Uranium and other isotopes | Concentration | (3) |
| **Examples of static + dynamic mechanism** | | | | | | | | | | | |
| Andrews et al. (2011) | Shale hill | Subsurface + Surface | Snowmelt and fall flushing periods | Low risk | Natural | Neutral | South-facing concave hillslopes | Snowmelt and fall flushing periods | Organic carbon | Concentration | (4) – (6) |
| Henri et al. (2015) | Preferential flow path | Subsurface | -- | High risk | Anthropogenic | Negative | Subsurface heterogeneity | Contaminant transport and travel time distribution | Chlorinated compounds | Concentration | (4) – (6) |
| Duncan et al. (2013) | Microtopography | Surface | Unimportant | High risk | Natural | Positive | Riparian hollows | Transport and retention of reactants | Nitrogen | Concentration or reaction rate | (4) – (6) |
| Arora et al. (2016) | Naturally reducing zone-induced transport | Subsurface | Temperature and water table fluctuation | Low risk | Anthropogenic + Natural | Neutral | Naturally reduced zones | Temperature and water table fluctuation | Carbon fluxes | Concentration or reaction rate | (4) – (6) |
| **Examples of multiple dynamic mechanisms** | | | | | | | | | | | |
| Hill et al. (2000) | Riparian zone | Subsurface | -- | High risk | Natural | Positive | Interfaces in the riparian zone | Supply of electron donor and acceptor from flow transport | Nitrogen and carbon | Concentration or reaction rate | (7) – (9) |
| Mitchell et al. (2008) | Peatlands | Subsurface + Surface | Summer periods | High risk | Natural | Negative | Upland-peatland interfaces induced by flow | Interactions between upland and peatland flow | Methylmercury | Concentration | (7) – (9) |
| Frei et al. (2012) | Microtopography | Surface | -- | Neutral | Natural | Neutral | Flowpaths induced by microtopography | Biogeochemical evolution along flow paths | Organic matter and nitrogen | Concentration or reaction rate | (7) – (9) |
| Gu et al. (2012) | Mixing zones | Subsurface + Surface | River discharge + Water table fluctuation | High risk | Natural | Positive | Mixing zones caused by river stages | Interaction between surface water and groundwater | Nitrogen | Concentration or reaction rate | (7) – (9) |


Table 1. Example cases considered in this study for constructing the statistical formulation of HSHM.





### 3.1 HSHMs triggered by static contributors only

In this section, we use Wainwright et al. (2015) as an example to illustrate our process to construct $I_{HSHM}(\boldsymbol{\Omega}^*, t^*)$ following Eq. (3), where an HSHM is triggered by static contributors only (section 2.1). NRZs within

floodplain environments at the Rifle site are considered biogeochemical hot spots because they represent elevated concentrations of uranium, organic matter, and geochemically reduced minerals and they have been found to contribute to significant carbon fluxes to the atmosphere and to local rivers (Arora et al., 2016). Due to its characteristics, we considered the spatial distribution of an NRZ to be a static-mechanism-based hot spot. Wainwright et al. (2015) used geophysical data (e.g., induced polarization) to map the distribution of an NRZ at the subsurface

level. They found that the phase shift ($\phi$) from the induced polarization data of the NRZ was within $[4.5, 5]mrad$, compared to non-NRZ locations at $\phi \subseteq [1, 3.5]mrad$. Thus, $\phi$ can be used to construct the static indicator with a critical condition of $[4.5, 5]mrad$. Therefore,

$$I_s(\boldsymbol{\Omega}^*) = \begin{cases} 1, & if \ Z_\phi(\boldsymbol{\Omega}^*) \subseteq [4.5, 5] \ mrad \\ 0, & otherwise \end{cases}. \tag{10}$$

Other static attributes, including but not limited to elevation, hydraulic conductivity, and resistivity, can also

be used to define the critical conditions to construct the static indicator for hot spots through Bayesian conditioning.

### 3.2 HSHMs occuring when dynamic contributors coincide at locations defined by static contributors

The second case we present here utilizes Eq. (4)–(6), where HSHMs are triggered when dynamic contributors coincide at hot spots determined by static contributors. Here, we present the case investigated by Duncan et al. (2013), where riparian hollows representing less than 1% of the total catchment area contributed to more than 99% of the total

denitrification within the watershed. In their study, the denitrification rates peaked during the base flow (midsummer) period, when the riparian hollows were partially oxygenated and the hydrologic fluxes were at a minimum. The site was considered to have low inorganic N availability, and thus, nitrate was supplied via nitrification. The highest rates of denitrification were therefore tied to nitrification and the partially aerated conditions.

The static indicator needs to be constructed based on the microtopographical features within the riparian

zone. Specifically, the topographic wetness index (TWI) (Beven and Kirkby, 1979; Sørensen et al., 2006) was used in Duncan et al. (2013) to delineate the riparian hollows from other riparian locations. Terrain analysis indicated a TWI threshold value of 6.0 and 8.0 for riparian hollows under wet and dry conditions, respectively, whereas 4.8 and smaller TWI values corresponded to other riparian locations (e.g., hummocks). Thus, the static indicator can be constructed using the TWI values within the riparian zone to determine the hot spot locations—the hollows. Hence,

$$I_s(\boldsymbol{\Omega}^*) = \begin{cases} 1, & if \ Z_{TWI}(\boldsymbol{\Omega}^*) > 6 \ (wet \ condition) \ or \ 8 \ (dry \ condition) \\ 0, & otherwise \end{cases}. \tag{11}$$

Multiple dynamic processes control the denitrification rate at the riparian hollows. As examined by Duncan et al. (2013), a partially aerated condition ($C_{O_2} > 5\%$) is needed to support nitrification, which supplies the nitrate for denitrification. As quiescent, non-storm periods during base flow favor the coupled nitrification-denitrification mechanism, this is another key process that needs to be represented by a dynamic indicator. Although Duncan et al.

(2013) did not mention specific concentration ranges for nitrogen species, the major components, such as organic N, should be available. Therefore, we can construct the dynamic indicators as follows:

$$P[I_d(\boldsymbol{\Omega}^*, t^*) = 1] = P[I_{d,O_2}(\boldsymbol{\Omega}^*, t^*) = 1, I_{d,Hydro}(\boldsymbol{\Omega}^*, t^*) = 1, I_{d,N}(\boldsymbol{\Omega}^*, t^*) = 1], \quad (12)$$

where $I_{d,Hydro}(\boldsymbol{\Omega}^*, t^*)$ is the dynamic indicator representing the streamflow stages; this will be 1 if the base flow conditions are met. Additionally, here, $I_{d,N}(\boldsymbol{\Omega}^*, t^*)$ is the dynamic indicator for the transport of the nitrogen species

in the subsurface that support the coupled nitrification-denitrification mechanism.

$$I_{d,O_2}(\boldsymbol{\Omega}^*, t^*) = \begin{cases} 1, & if\ C_{O_2}(\boldsymbol{\Omega}^*, t^*) > 5\% \\ 0, & otherwise \end{cases},$$

$$I_{d,Hydro}(\boldsymbol{\Omega}^*, t^*) = \begin{cases} 1, & if\ t^* \subseteq base\ flow\ periods \\ 0, & otherwise \end{cases}, \quad (13)$$

$$I_{d,N}(\boldsymbol{\Omega}^*, t^*) = \begin{cases} 1, & if\ C_N(\boldsymbol{\Omega}^*, t^*) > 0 \\ 0, & otherwise \end{cases}.$$

It is noted that these dynamic processes are not statistically independent. Usually, when one condition is met
(e.g., base flow conditions), other conditions may consistently be satisfied (e.g., the transport of nitrogen in riparian hollows). Alternatively, numerical modeling approaches are more feasible to construct the dynamic indicators based on the critical conditions at riparian hollows ($\boldsymbol{\Omega}^*$), where we could directly target $N_2$ fluxes using a Monte Carlo approach. The statistical formulation used here is constructed specifically for the mechanisms described by Duncan et al. (2013). Thus, the detailed threshold limits could change under other denitrification HSHMs cases, such as the
case presented in Hill et al. (2000), who focus on desert landscapes, or the one by Harms and Grimm (2008), where the monsoon season is influential for the nitrogen transport. Nonetheless, the general formulation of HSHMs using indicators is still applicable.

### 3.3 HSHMs occuring when multiple dynamic processes converge in space

HSHMs can also be triggered by the confluence of multiple dynamic processes that lead to the convergence
of complementary reactants at $\boldsymbol{\Omega}^*$. Accumulation of complementary reactants is mobilized and transported via different hydrologic flowpaths. They converge at hot spot locations and trigger hot moments during the mixing. Following the statistical framework developed in this study, Eq. (7) to (9) are suitable for this condition. In order to illustrate how the dynamic indicators are constructed, we consider here the case reported by Gu et al. (2012), where high biogeochemical activity was observed at the interface of groundwater and surface water during the stream stage
fluctuations, which resulted in significant in-stream denitrification and $NO_3^-$ removal.

In their study, hot spots form around the near-stream-riparian subsurface during river stage fluctuations, where active biogeochemical reaction (e.g., denitrification) requires both $O_2$ depletion and the simultaneous presence of $NO_3^-$ and the dissolved organic carbon (DOC). Specifically, the spatiotemporal distribution of denitrification hot spots coincides with an $O_2$ depletion zone along the DOC infiltration flowpaths. In order to determine the mixing of
groundwater and surface water during stage fluctuations, Gu et al. (2012) defined bank storage volume $V(t)$ and maximum bank storage volume $V_{max}$. The flood hydrograph was subdivided into the rising limbs, recession limbs and return flow, the latter representing the slow restitution of part of the water that infiltrated during the previous stages. Considering the different dynamics of these components, they observed that the largest infiltration rate occurred prior



to the maximum stage rise, while $V_{max} = 5m^3m^{-1}$ (critical condition) occurred in the recession limb of the flood

event. Instead, maximum return flow occurred toward the end of the recession curve before stream hydrograph

stabilizes. Maximum $NO_3^-$ rate removal occurred when return flow phase was almost complete and then decreased

until the depletion of $NO_3^-$. Through statistical analysis, they found that $V_{max}$, viewed as an integrated index for

hydrological exchange, could explain 64% of the variation in the $NO_3^-$ removal. Thus, $V_{max}$ can be used as the critical

state to determine whether or not the hyporheic dynamics is significant to enhance relevant biogeochemical processes.

In order for the hot moments to be significant, the stream-riparian zone should also be microbially active. Based on

these conditions, the dynamic indicators can be constructed as follows:

$$P[I_d(\boldsymbol{\Omega}^*, t^*) = 1] = P[I_{d,Hydro}(\boldsymbol{\Omega}^*, t^*) = 1, I_{d,Chem}(\boldsymbol{\Omega}^*, t^*) = 1], \tag{14}$$

where $I_{d,Hydro}(\boldsymbol{\Omega}^*, t^*)$ represents the dynamic process induced by the hydrologic conditions (e.g., stage fluctuation),

and $I_{d,Chem}(\boldsymbol{\Omega}^*, t^*)$ represents the dynamic process controlled by the transport and accumulation of chemical reactants.

Based on the critical values or ranges, we formulate the indicators as follows:

$$I_{d,Hydro}(\boldsymbol{\Omega}^*, t^*) = \begin{cases} 1, & Z_{V_{max}}(\boldsymbol{\Omega}^*, t^*) \geq 5m^3m^{-1} \\ 0, & otherwise \end{cases},$$

$$I_{d,Chem}(\boldsymbol{\Omega}^*, t^*) = \begin{cases} 1, & if\ C_{O_2}(\boldsymbol{\Omega}^*, t^*)\ is\ small\ and\ C_{NO_3^-}(\boldsymbol{\Omega}^*, t^*) > 0\ and\ C_{DOC}(\boldsymbol{\Omega}^*, t^*) > 0 \\ 0, & otherwise \end{cases}. \tag{15}$$

Typically, because of the complexity of the processes, no analytical solutions are available for formulating

the indicators. However, Monte Carlo simulations can be useful in constructing such indicators. For this case, an

HSHM at any given location and time $(\boldsymbol{\Omega}^*, t^*)$ will only be triggered when all of the conditions are met and the

ensemble mean of the indicator assumes the following form:

$$< I_d(\boldsymbol{\Omega}^*, t^*) > = \frac{1}{N} \sum_{i=1}^{N} I_{d,i}(\boldsymbol{\Omega}^*, t^*), \tag{16}$$

where $I_{d,i}(\boldsymbol{\Omega}^*, t^*)$ is the value that the indicator assumes in the $i^{th}$ realization and $N$ is the total number of simulations.

Overall, our choices of the three studies should not limit the generalizability of the indicator statistics

approach for deriving statistical formulations for HSHM applications. The critical conditions chosen to construct the

indicators are determined solely on the findings from these selected studies, and they will vary under different

scenarios.

## 4 HSHM applications in groundwater hydrology

Processes occurring within the subsurface are important factors leading to HSHM occurrences. Among

others, these processes include the migration of groundwater carrying reducing substrates, nuclear waste transport

within the subsurface, the accumulation and transport of dense non-aqueous phase liquid (DNAPL) and other

biogeochemical processes. Some current modeling approaches that focus on subsurface HSHMs assume simplified

hydrologic structures (e.g., homogeneous and isotropic domains) in quantifying contaminant fate and transport in the

subsurface. However, such an assumption neglects the effect of the heterogeneity in the subsurface, leading to the

underestimation of the uncertainties in the HSHM occurrences. Thus, in this section, we focus on HSHM applications

in groundwater hydrology, with a particular emphasis on spatial variability in the subsurface. Specifically, we consider





several situations often encountered in groundwater contamination studies and present the indicator statistical formulations of HSHMs. With these results, we can determine the probability of HSHMs occurrences in the subsurface at a given time and space. Further, we are able to determine how spatial variability influences HSHM occurrences and how this is translated into environmental health risks.

### 4.1 Importance of spatial variability in the subsurface

The heterogeneous structure of hydraulic conductivity leads to significant variability in the contaminant transport in the subsurface, which further results in the heterogeneity of biogeochemical cycling, such as the development of NRZs, reactive facies, and heterogeneity in aquifers' reactivity (Li et al., 2010; Loschko et al., 2016; Sassen et al., 2012; Wainwright et al., 2015).

Figure 2 demonstrates the uncertainty associated with HSHMs by looking at the flow fields in two-dimensional log-hydraulic conductivity ($Y = ln(K)$) fields with streamlines resulting from a uniform mean head gradient, left to right. The three panels differ in terms of the variance, $\sigma_Y^2$, of the log-conductivity. The covariance function used for generating the fields is exponential and isotropic. $\sigma_Y^2$ is shown to have a profound impact upon the conductivity field. As the variance increases, regions of high and low log-conductivity emerge, creating preferential flow paths bypassing the low conductivity zones as shown by particle trajectories. At smaller variance (i.e., $\sigma_Y^2 = 0.1$), particles mainly travel along the mean flow direction with very limited departure from the mean trajectory, which are the straight lines connecting the left and right boundaries. In this situation, the arrival times of solute particles to critical locations (i.e., $\boldsymbol{\Omega}^*$) are predictable. With large variances (i.e., $\sigma_Y^2 = 2$), the streamlines assume a very irregular, hard-to-predict geometry, and we can observe the emergence of flow channels, where particles can move fast, next to stagnant flow regions. Arrival times become more uncertain, because the exact geometry of the streamlines is hard to predict unless the $Y$ field is known deterministically. However, in another realization of the $Y$ field, the situation may be totally different, resulting in significant uncertainties in predicting the particle travel times. Thus, spatial variability of log-conductivity is a major uncertainty-inducing factor, and by extension, obviating the need for stochastic modeling of HSHMs in situations where the associated processes and attributes are subject to uncertainty. In the following sections, we will present illustrative examples to analyze how subsurface spatial variability influences $< I_{HSHM}(\boldsymbol{\Omega}^*, t^*) >$, including variance and anisotropy ratio of the log-conductivity.

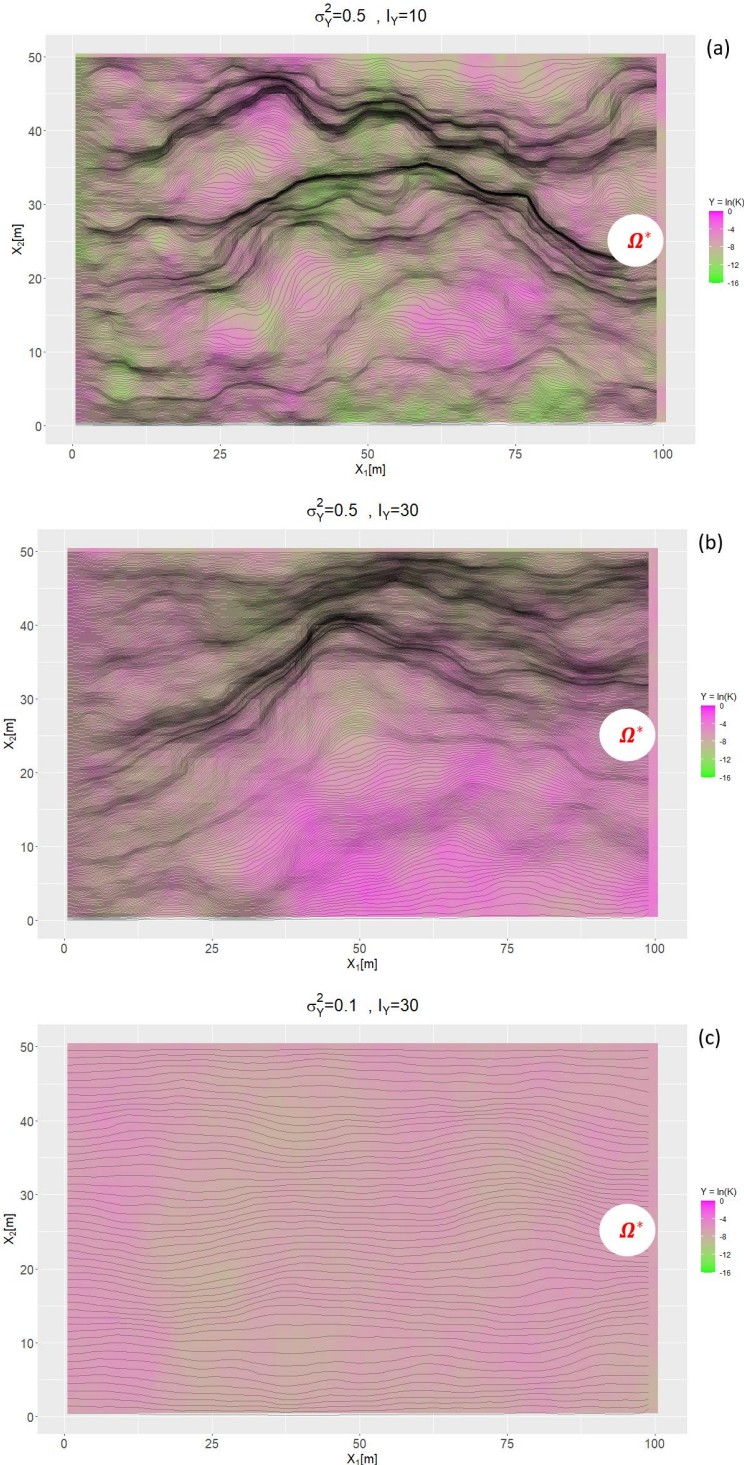





Figure 2. Illustrative example of a heterogeneous log-hydraulic conductivity field and solute particle transport. Black lines represent simulated particle travel paths. A left to right hydraulic gradient of 0.1 is applied. Mean of log-conductivity is set at -3. Note color scales for log-conductivity are consistent in all three panels.

### 4.2 Case studies and expansions of indicators

#### 4.2.1 Single-particle $I_d$ within $\boldsymbol{\Omega}^*$

Consider the case of a point source release of non-reactive tracer originated from $(\boldsymbol{x_0}, t_0)$. The dynamic indicator depends on a particle being within $\boldsymbol{\Omega}^*$ at time $t^*$ or not. If local (pore scale) dispersion is neglected, the dynamic indicator is defined as follows:

$$I_d(\boldsymbol{\Omega}^*, t^*) = \begin{cases} 1, & if\ \boldsymbol{X}(t^*) \subseteq \boldsymbol{\Omega}^*\ at\ t^* \\ 0, & otherwise \end{cases}. \tag{17}$$

Given that the particle does not change its volume while traveling. The expected value of this dynamic indicator at $t^*$ is therefore:

$$< I_d(\boldsymbol{\Omega}^*, t^*) > = \int_{\Omega^*} f_{X(t^*)}(\boldsymbol{a}|\boldsymbol{x}_0, t_0)d\boldsymbol{a}, \tag{18}$$

where $f_{X(t^*)}(\boldsymbol{a}|\boldsymbol{x}_0, t_0)$ is the probability distribution function (pdf) of the particle's trajectory at $t^*$ (Dagan and Nguyen, 1989; Rubin, 2003). Other situations may be addressed by using the same framework. For example, for an instantaneous injection within a source volume $V_0$, the ensemble mean of the dynamic indicator assumes the following form:

$$< I_d(\boldsymbol{\Omega}^*, t^*) > = \frac{1}{V_0} \int_{V_0} \int_{\Omega^*} f_{X(t^*)}(\boldsymbol{a}|\boldsymbol{x}_0, t_0)d\boldsymbol{a}d\boldsymbol{x}_0. \tag{19}$$

#### 4.2.2 Concentration-based $I_d$ within $\boldsymbol{\Omega}^*$

When considering local dispersion, or in case of a reactive tracer, the condition that the particle is inside the volume $\boldsymbol{\Omega}^*$ does not suffice to define the dynamic indicator and a concentration threshold $C_{th}$ should be introduced:

$$I_d(\boldsymbol{\Omega}^*, t^*) = \begin{cases} 1, & if\ \boldsymbol{X}(t^*; \boldsymbol{x}_0, t_0) \subseteq \boldsymbol{\Omega}^*\ and\ C(\boldsymbol{X}, t^*) > C_{th} \\ 0, & otherwise \end{cases}. \tag{20}$$

In the absence of local dispersion and for a reactive solute decaying at a (spatially) constant rate $k$, the ensemble mean assumes the following expression (Cvetkovic and Shapiro, 1990):

$$< I_d(\boldsymbol{\Omega}^*, t^*) > = \left\{1 - H\left[t^* - \frac{1}{k}\ln(\frac{C_0}{C_{th}})\right]\right\} \int_{\Omega^*} f_{X(t^*)}(\boldsymbol{a}|\boldsymbol{x}_0, t_0)d\boldsymbol{a}, \tag{21}$$

where $C_0$ is the initial concentration and $H[\cdot]$ is the Heaviside step function. The ensemble mean (21) is the product of the probability that the particle assumes a concentration larger than the threshold at $t^*$ (given that reaction rate $k$ is constant, this probability is either 0 or 1) and the probability that at the same time $t^*$ the particle is within the hot spot $\boldsymbol{\Omega}^*$. In other words, Eq. (21) expresses the fact that a particle inside $\boldsymbol{\Omega}^*$ contributes to the hot moment only if its concentration is greater than the threshold. Equation (21) can be generalized to the cases of instantaneous injection into a source of volume $V_0$, as discussed before for the non-reactive case. For other complex situations, such as that in which $k$ is spatially variable and complex reaction networks, the ensemble mean of the indicators can be addressed by Eq. (16) in a Monte Carlo framework.



### 4.2.3 Assessing the duration of hot moment and probabilities

The probability that the hot moment persists over the interval $[t_1, t_2]$ at $\boldsymbol{\Omega}^*$ can be formally computed as follows:

$$< I_d(\boldsymbol{\Omega}^*, t_1, t_2) >= P(t_1, \boldsymbol{\Omega}^*) P(t_2 | t_1, \boldsymbol{\Omega}^*), \tag{22}$$

where $P(t_1, \boldsymbol{\Omega}^*)$ is the probability that the particle is inside $\boldsymbol{\Omega}^*$ at time $t^* = t_1$ and $P(t_2 | t_1, \boldsymbol{\Omega}^*)$ is the probability that the particle is still inside $\boldsymbol{\Omega}^*$ at time $t^* = t_2$, provided that at time $t_1$, it was also inside $\boldsymbol{\Omega}^*$. If the particle exits $\boldsymbol{\Omega}^*$ during interval $[t_1, t_2]$, this time interval will not be qualified as hot moment; and thus the probability computation needs to ensure the particle stays within $\boldsymbol{\Omega}^*$ during the entire time interval.

Under the First-Order Approximation (FOA) (see e.g., Dagan, 1989; Gelhar 1993; Rubin, 2003), the pdf of
the particle displacement is normal with mean $< X(t^*; \boldsymbol{x_0}, t_0) >$ and auto-covariance tensor of the residual displacements $\boldsymbol{X}'(t^*) = \boldsymbol{X}(t^*) - \langle \boldsymbol{X}(t^*) \rangle$ defined by $X_{ij}(t^*; \boldsymbol{x_0}, t_0) = \langle X'_i(t^*; \boldsymbol{x_0}, t_0) X'_j(t^*; \boldsymbol{x_0}, t_0) \rangle, \; i, j = 1, 2, 3.$ For simplicity in the following, we assume $\boldsymbol{x_0} = 0 \; and \; t_0 = 0.$ Under these assumptions,

$$< I_d(\boldsymbol{\Omega}^*, t_1, t_2) >= \int_{\boldsymbol{\Omega}^*} \int_{\boldsymbol{\Omega}^*} f_{X(t_1)}(\boldsymbol{a}) f^c_{X(t_2)}(\boldsymbol{b} | X(t_1) = \boldsymbol{a}) \, d\boldsymbol{b} \, d\boldsymbol{a}, \tag{23}$$

where the conditional pdf $f^c_{X(t_2)}(\boldsymbol{b} | X(t_1) = \boldsymbol{a})$ is multi-normally distributed with conditional mean and variance
tensor given by

$$\langle X(t_2) | X(t_1) = \boldsymbol{a} \rangle = < X(t_2) >$$
$$+ Cov[\boldsymbol{X}'(t_2), \boldsymbol{X}'(t_1)] \cdot Var[\boldsymbol{X}'(t_1)]^{-1} \cdot (\boldsymbol{a} - < X(t_1) >), \tag{24}$$

and

$$\boldsymbol{\sigma}(t_1, t_2) = Var[\boldsymbol{X}'(t_2)] - Cov[\boldsymbol{X}'(t_2), \boldsymbol{X}'(t_1)] \cdot Var[\boldsymbol{X}'(t_1)]^{-1} \cdot Cov[\boldsymbol{X}'(t_1), \boldsymbol{X}'(t_2)], \tag{25}$$

respectively, which further yields the following,

$$f^c_{X(t_2)}(\boldsymbol{b} | X(t_1) = \boldsymbol{a})$$
$$= \exp\left[ -\frac{1}{2} \left[\boldsymbol{b} - \langle X(t_2) | X(t_1) = \boldsymbol{a} \rangle\right]^T \cdot \boldsymbol{\sigma}(t_1, t_2)^{-1} \cdot \left[\boldsymbol{b} - \langle X(t_2) | X(t_1) = \boldsymbol{a} \rangle\right] \right]$$
$$\cdot \{8 \, \pi^3 \cdot |\boldsymbol{\sigma}(t_1, t_2)|\}^{-\frac{1}{2}}, \tag{26}$$

where $|\cdot|$ indicates the determinant, $exp$ is the exponential function and the exponent T indicates the transpose of the
vector.

In Eq. (24) and (25), $\boldsymbol{X}'(t^*) = \boldsymbol{X}(t^*) - \langle \boldsymbol{X}(t^*) \rangle$ stands for the departure of the particle's displacement with respect to the ensemble mean trajectory, and $Var[\boldsymbol{X}]^{-1}$ is the auto-covariance tensor of the residual displacement whose elements are defined above. Similarly, $Cov[\boldsymbol{X}'(t_1), \boldsymbol{X}'(t_2)]$ is the covariance tensor of residual displacement which elements are: $X_{ij}(t_1, t_2; \boldsymbol{x_0}, t_0) = \langle X'_i(t_1) X'_j(t_2) \rangle, i, j = 1, 2, 3.$ Note that in the general three-dimensional case
$\langle \boldsymbol{X}(t_2) | \boldsymbol{X}(t_1) = \boldsymbol{a} \rangle$ is a three-dimensional vector and $\boldsymbol{\sigma}(t_1, t_2)$ is a $3 \times 3$ second-order tensor.

For $t_2 \to t_1$, $f_{X(t_2)}[\boldsymbol{b} | X(t_1) = \boldsymbol{a}] \to \delta(\boldsymbol{b})$, where $\delta(\cdot)$ is the Dirac Delta, such that $P(t_2 | t_1, \boldsymbol{\Omega}^*) \to 1$. On the other hand, for $t_2 \gg t_1$, $Cov[\boldsymbol{X}'(t_1), \boldsymbol{X}'(t_2)] \to 0$ and $P(t_2 | t_1, \boldsymbol{\Omega}^*) \to P(t_2, \boldsymbol{\Omega}^*)$ the marginal probability that the particle is within $\boldsymbol{\Omega}^*$ at time $t^* = t_2$. Equations (23) to (26) are obtained under the FOA approximation and assuming that the particle can enter $\boldsymbol{\Omega}^*$ only once. Such assumption is needed to obtain analytical solutions and is reasonable for
situations with small to mild subsurface heterogeneity (e.g., $\sigma_Y^2 \leq 1.6$), such as the cases presented in Bellin et al.





(1992, 1994); Cvetkovic et al. (1992). In particular, FOA assumes small heterogeneity and under this assumption the particle trajectory deviates slightly from its ensemble mean, which is directed along the regional hydraulic head gradient. For a regular volume $\boldsymbol{\Omega}^*$, this reduces the probability of the particle entering more than once the hot spot. This probability reduces further if in horizontal and vertical transverse directions $\boldsymbol{\Omega}^*$ is much larger than the respective

integral scales, because the probability of observing negative longitudinal velocity components (i.e., along the mean flow field) is much smaller than in the transverse directions (Bellin et al., 1992) and vanishes as formation heterogeneity reduces.

If the hotspot $\boldsymbol{\Omega}^*$ is the volume confined between two planes at $x_1 - \frac{l_1}{2}$ and $x_1 + \frac{l_1}{2}$, with the other two dimensions much larger than the transverse horizontal and vertical integral scales: $l_2 \gg I_h$, $l_3 \gg I_v$, Eq. (24) simplifies

to:

$$< I_d(\boldsymbol{\Omega}^*, t_1, t_2) > = \int_{x_1 - \frac{l_1}{2}}^{x_1 + \frac{l_1}{2}} \int_{x_1 - \frac{l_1}{2}}^{x_1 + \frac{l_1}{2}} f_{X_1(t^*)}(a_1) f^c_{X_1(t^*)}(b_1 | X_1(t_1) = a_1) \, db_1 \, da_1, \qquad (27)$$

where $X_1$ is the longitudinal component of the particle's trajectory and $f^c_{X_1(t^*)}$ is its conditional pdf, which is normal with conditional mean and variance given by

$$\mu[a_1] = \langle X_1(t_2) | X_1(t_1) = a_1 \rangle = \, < X_1(t_2) > + \frac{X_{11}(t_1, t_2)}{X_{11}(t_1)} \, (X_1(t_1) - < X_1(t_1) >), \qquad (28)$$

and

$$\sigma^2(t_1, t_2) = X_{11}(t_2) - \frac{X_{11}(t_1, t_2)^2}{X_{11}(t_1)}, \qquad (29)$$

respectively. Consequently, $f_{X^c(t^*)}$ in Eq. (27) assumes the following form:

$$f^c_{X_1(t^*)}(b_1 | X_1(t_1) = a_1) = \frac{1}{\sqrt{2 \pi \, \sigma(t_1, t_2)}} \exp\left[ -\frac{1}{2} (b_1 - \mu[a_1])^2 \, \sigma(t_1, t_2)^{-1} \right]. \qquad (30)$$

Substituting Eq. (30) into Eq. (27) allows us to compute $< I_d(\boldsymbol{\Omega}^*, t_1, t_2) >$. For situations where the FOA assumptions

are not valid (e.g., large heterogeneity), Monte Carlo simulation framework is still applicable as alternative approach to construct the dynamics indicators (see Eq. 16).

### 4.3 Illustrative example and indicator formulation

Following sections 4.1 and 4.2, we present here synthetic case studies that demonstrate the statistical formulation of the indicators using methods developed in stochastic hydrogeology. The choice of the synthetic case

studies does not limit our approaches to broader applications where stochastic modeling with Monte Carlo simulations are applicable. In most applications, the locations of hot spots ($\boldsymbol{\Omega}^*$) are determined by static indicators, such as riparian hollows (Duncan et al., 2013), reactive facies (Sassen et al., 2012), and NRZs (Wainwright et al., 2015). The static indicator is constructed according to the corresponding critical conditions provided by ancillary data such as topography, remote sensing, and/or geophysical data. Hence, in this case, assuming the boundaries of $\boldsymbol{\Omega}^*$ are

determined by a static indicator, we consider a hot spot ($\boldsymbol{\Omega}^*$) to be confined within the following volume: $w_1 \leq x_1 \leq w'_1; w_2 \leq x_2 \leq w'_2; w_3 \leq x_3 \leq w'_3$.





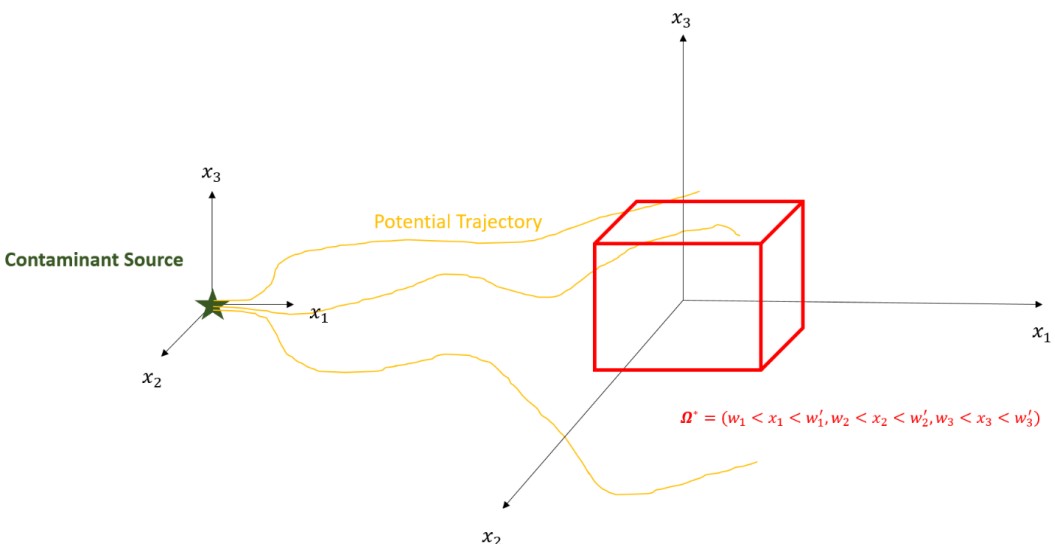

Figure 3. Configuration of the synthetic case study

Given this case, the hot moment will be triggered only when the contaminant particle is found within $\boldsymbol{\Omega}^*$. The

probability of finding the contaminant particle within $\boldsymbol{\Omega}^*$ is given by

$$prob\{X(t^*) \subseteq \boldsymbol{\Omega}^*\}$$

$$= \prod_{i=1}^{m} prob\{w_i \leq X_i(t^*) \leq w_i'\} = \prod_{i=1}^{m} \int_{w_i}^{w_i'} f_{X_i(t^*)}(a_i|x_0,t_0)da_i, \qquad (33)$$

where $m$ denotes the space dimensionality. Equation (33) defines the dynamic indicator for this case. If $\boldsymbol{\Omega}^*$ is already

identified as a hot spot location, then Eq. (33) provides $< I_{HSHM}(\boldsymbol{\Omega}^*,t^*) >$. Otherwise, the static indicator should be

incorporated to determine the boundaries of $\boldsymbol{\Omega}^*$ in order to compute $< I_{HSHM}(\boldsymbol{\Omega}^*,t^*) >$ as shown in Eq. (10) where

geophysical data is used to identify the spatial context of $\boldsymbol{\Omega}^*$. If we also assume steady, uniform in the average flow

with mild heterogeneity of the log hydraulic conductivity field with Gaussian displacement pdf—then we can compute

$< I_{HSHM}(\boldsymbol{\Omega}^*,t^*) >$ analytically using the following equation:

$$< I_{HSHM}(\boldsymbol{\Omega}^*,t^*) >=< I_s(\boldsymbol{\Omega}^*) > < (I_d(\boldsymbol{\Omega}^*,t^*) >$$


$$= prob(I_d(\boldsymbol{\Omega}^*,t^*)=1) = prob\{X(t^*) \subseteq \boldsymbol{\Omega}^*\}$$

$$= \prod_{i=1}^{m} \int_{w_i}^{w_i'} f_{X_i(t^*)}(a_i|x_0,t_0)da_i = \int_{w_1}^{w_1'} f_{X_1(t^*)}(a_1|x_0,t_0)da_1 \int_{w_2}^{w_2'} f_{X_2(t^*)}(a_2|x_0,t_0)da_2 \int_{w_3}^{w_3'} f_{X_3(t^*)}(a_3|x_0,t_0)da_3$$

$$= \frac{1}{(2\pi)^{\frac{3}{2}}\sqrt{X_{11}(t^*)X_{22}(t^*)X_{33}(t^*)}} \int_{w_1}^{w_1'} \exp\left[-\frac{1}{2}\frac{(a_1-Ut^*)^2}{X_{11}(t^*)}\right]da_1$$

$$\cdot \int_{w_2}^{w_2'} \exp\left[-\frac{1}{2}\frac{a_2^2}{X_{22}(t^*)}\right]da_2 \int_{w_3}^{w_3'} \exp\left[-\frac{1}{2}\frac{a_3^2}{X_{33}(t^*)}\right]da_3. \qquad (34)$$





which can be integrated to yield :

$$< I_{HSHM}(\boldsymbol{\Omega}^*, t^*) > = \frac{1}{8}\left[\text{erfc}\left(\frac{w_1 - Ut^*}{\sqrt{2X_{11}(t^*)}}\right) - \text{erfc}\left(\frac{w_1' - Ut^*}{\sqrt{2X_{11}(t^*)}}\right)\right]$$

$$\cdot\left[\text{erfc}\left(\frac{w_2}{\sqrt{2X_{22}(t^*)}}\right) - \text{erfc}\left(\frac{w_2'}{\sqrt{2X_{22}(t^*)}}\right)\right]\left[\text{erfc}\left(\frac{w_3}{\sqrt{2X_{33}(t^*)}}\right) - \text{erfc}\left(\frac{w_3'}{\sqrt{2X_{33}(t^*)}}\right)\right]. \quad (35)$$

The form of the displacement variances is controlled by the spatial distribution of the hydraulic conductivity in the subsurface. Equations (A4)-(A6) of the appendix show the displacement variances for an axisymmetric exponential covariance function of the log-conductivity (A3).

### 4.4 Implications for HSHMs

In the following sections, we present the results from the case study described in section 4.3. Specifically, in section 4.4.1 and 4.4.2, we explore how heterogeneity of log-hydraulic conductivity influences the probability of HSHM occurrences. To make results as general as possible, lengths are made dimensionless with respect to the integral scales ($I_{Yh}$ in the two horizontal directions and $I_{Yv}$ in the vertical one) and time with respect to the following advective time scale: $I_{Yh}/U$, where $U$ is the mean velocity). In the following, we explore the effect of the remaining parameters, i.e. the anisotropy ratio $e = \frac{I_{YV}}{I_{YH}}$ and the variance of the log-conductivity $\sigma_Y^2$, on the emergence of HSHM. We placed $\boldsymbol{\Omega}^*$ along the mean trajectory at $(21I_{YH}, 0, 0)$ with dimensions as $(2I_{YH}, 2I_{YH}, 2I_{YV})$. The dimensions of the hot spot are therefore of two integral scales in the three coordinate directions $(x_1, x_2, x_3)$ and is placed at a dimensionless distance of 21 from the point source.

### 4.4.1 Dependece of $< I_{HSHM}(\boldsymbol{\Omega}^*, \tau) >$ on variance in the spatial correlation structure of the log-conductivity

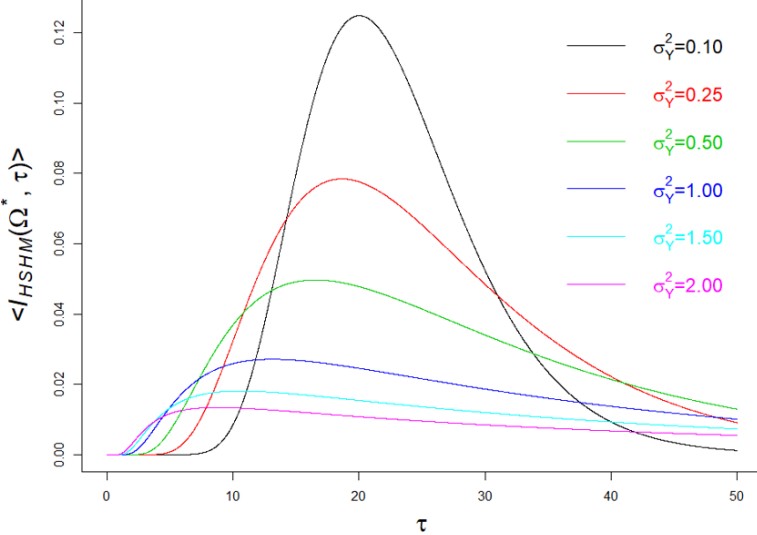

Figure 4. Dependence of $< I_{HSHM}(\Omega^*, \tau) >$ on $\sigma_Y^2$



Isotropic heterogeneity ($e = 1$ and the particle moments given by Eqs. (A7) and (A8)) was considered to investigate the dependence of $< I_{HSHM}(\boldsymbol{\Omega}^*, \tau) >$ on $\sigma_Y^2$ with results presented in Figure 4. $\tau = tU/I_{Yh}$ is the dimensionless time. At early time (e.g., $\tau < 5$), larger probability $< I_{HSHM}(\boldsymbol{\Omega}^*, \tau) >$ is observed with increase in $\sigma_Y^2$. At intermediate time, i.e., at times comparable with the mean travel time $\tau = 21$, $< I_{HSHM}(\boldsymbol{\Omega}^*, \tau) >$ is inversely proportional to $\sigma_Y^2$. At late time (e.g., $\tau > 40$), the largest $< I_{HSHM}(\boldsymbol{\Omega}^*, \tau) >$ occurs at intermediate $\sigma_Y^2$. We observe that $\sigma_Y^2$ regulates the timing of the peak in $< I_{HSHM}(\boldsymbol{\Omega}^*, \tau) >$, which is located in the proximity of the mean travel time, $\tau = 21$, for weak heterogeneity, and shifts towards earlier times as $\sigma_Y^2$ increases.

These effects relate to the relationship between travel times (from the source to $\boldsymbol{\Omega}^*$) and $\sigma_Y^2$. The key point to note is that $\sigma_Y^2$ controls the spread of the travel time around the mean travel time. Larger variance enhance channeling effects (Fiori and Jankovic, 2012; Moreno and Tsang, 1994, also in Figue 2), which in turn enable earlier arrival times. But at the same time, large $\sigma_Y^2$ also leads to the low-conductivity zones. Streamlines of the solute tend to bypass low hydraulic conductivity zones, however, the small amount of solute that actually penetrates these zones by advection and diffusion gets trapped for long time before being released and this results in an extended tailing with low concentration and therefore low $< I_{HSHM}(\boldsymbol{\Omega}^*, \tau) >$. Thus, with an increase in $\sigma_Y^2$, we notice an increase in the probability to observe both increasingly earlier and increasingly delayed arrival times, which widens the probability distribution. On the contrary at small variance, particles deviate little from the ensemble mean trajectory, because of the small contrast in conductivity between high and low conductivity zones. This results in small particle spreading and travel times that differ only slightly from the mean travel time ($\tau = 21$), and a probability distribution less spread around the mean, where the peak is observed.

In summary, hydraulic conductivity contrast between low and high conductive lithofacies increases with $\sigma_Y^2$ leading to the emergence of organized high conductivity pathways sneaking through surrounding low conductivity zones with the latter acting as "trapping" elements. This causes the emergence of both early and late arrival times. Early arrival times are controlled by the connected high conductivity pathways and the late arrival times are influenced by the low conductivity zones, which act as low-release reservoirs for solutes.





### 4.4.1 Dependece of $< I_{HSHM}(\Omega^*, \tau) >$ on on anisotropy in the spatial correlation structure of the log-hydraulic conductivity

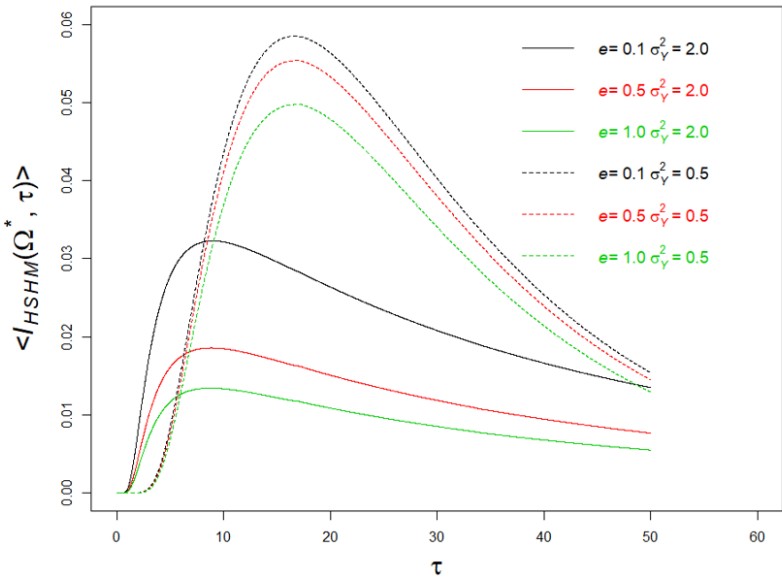


Figure 5. Dependence of $< I_{HSHM}(\Omega^*, \tau) >$ on $e$

The discussion here (accompanying Figure 5) focuses on the impact of the anisotropy ratio in the correlation structure ($e$, defined above) on the HSHM probabilities. The anisotropy ratio, $e$, provides an indication about the persistence of the log-conductivity ($Y$) in the various directions. The spatial correlation model used here for

demonstration is that of axis-symmetry, which is common to sedimentary formations (Dagan, 1989; Rubin, 2003), with $e$ providing the ratio between the persistence of $Y$ in the vertical ($x_3$) direction, represented by $I_{YV}$, and the ones on the horizontal plane ($x_1 - x_2$), represented by $I_{YH}$. In unconsolidated sedimentary formations, $I_{YV}$ is typically smaller than $I_{YH}$ by as much as one order of magnitude, due to the different time scales of the depositional process taking place in the horizontal and vertical directions, which leads to thin and elongated lithofacies and consequently

to a much smaller persistence of $Y$ values in normal to the horizontal plane (Miall, 1985, 1988; Ritzi et al., 2004).

Figure 5 compares $< I_{HSHM}(\Omega^*, \tau) >$ between formations defined by different anisotropy ratios and different $\sigma_Y^2$. It shows that we have two factors to consider when explaining the differences in $< I_{HSHM}(\Omega^*, \tau) >$. First factor, as discussed earlier, is the expansion in the range of travel times due to increase in $\sigma_Y^2$. With larger variance, we observe higher probabilities for departure of the travel times away from the average. The anisotropy ratio $e$ adds a

compounding factor. To understand its effect, we should recall the analyses of lateral displacement variances of solute particles moving in heterogeneous formations (cf., Dagan, 1989, and Eq. A4 to A6 here), showing that smaller $e$ leads to smaller lateral (both vertical and horizontal) displacement variances, implying smaller probabilities for lateral departures from the mean flow trajectory. Smaller $e$ limits lateral spreads, and increase the probability of particle to enter $\Omega^*$ , sooner or later, and to trigger HSHM. The effect could also be viewed as a channeling effect of sorts:





565 smaller $e$ implies $Y$ blocks of small aspect ratio (i.e., long but thin elements), which provide fast tracks for particles when defined by high $Y$ values, while blocking lateral spreads when defined by low $Y$ values.

  There are a few additional things to note here for completeness. First, $\boldsymbol{\Omega}^*$ in the present analysis is located downstream from the source, along with the mean trajectory of the solute displacement. We expect different results in situations where $\boldsymbol{\Omega}^*$ is positioned at an offset with respect to the mean flow direction. Second, we note that the

570 analytical models used to compute the displacement statistics are formally limited to smaller variance ($\sigma_Y^2 < 1$), although they are shown to provide good approximations for large variances (Bellin et al., 1992).Third, the stochastic formulation provides the theoretical and computational formalism for conditioning the  probabilities on in-situ measurements (Copty et al., 1993; Ezzedine and Rubin, 1996; Hubbard et al., 1997; Maxwell et al., 1999; Rubin, 1991a; Rubin et al., 1992; Rubin and Dagan, 1992) as well as on information  borrowed from similar sites (Li et al.,

575 2018; Cucchi et al., 2019).

**5 Discussion and Summary**

  In this study, we developed a general stochastic framework that could be used to characterize the spatiotemporal distribution of environmental Hot Spots Hot Moments (HSHMs), with groundwater applications. The stochastic formulation is built around the following principles:

580 • The HSHMs are defined as random variables and the goal is to derive their stochastic distribution in terms of the relevant processes and attributes.

• HSHMs processes cover the dynamic components of the HSHMs. An example could be the transport of solutes and reactants. HSHMs attributes refer to the static components of the HSHMs, e.g., in situations related to the nitrogen cycles, attributes could represent pyrite concentrations or naturally-reducing zones.

585 HSHMs could be defined through the confluence of a variety of contributors, both static and dynamic.

• The processes and attributes are modeled as stochastic processes and random variables, respectively, based on the underlying physics.

• The static contributors are modeled stochastically using geostatistical space random functions.

• The dynamic contributors are modeled stochastically using probability distribution functions derived from

590 the underlying mathematical-physical models.

• Several HSHMs categories are defined, based on the contributing factors, as follows: HSHMs defined by dynamic contributors only, HSHMs defined by static contributors, and most commonly, HSHMs requiring the coupling of static and dynamic contributors. The HSHMs stochastic formulations are expressed in terms of the stochastic formulations of the relevant contributors.

595 • We provided a detailed review of multiple HSHMs and showed how they relate to our definitions.

  The framework we proposed in this study is advantageous in that it allows to calculate the uncertainty associated with HSHMs based on the uncertainty associated with its contributors. Additionally, it provides a formalism, well established by Bayesian theory, for conditioning the HSHM probabilities on *in-situ* measurements as well as on information borrowed from geologically and otherwise similar sites.





We demonstrated our proposed approach through applications in the area of subsurface transport and hydrogeology, focusing on the impacts of subsurface heterogeneity on HSHMs. We analyzed, quantitatively, how subsurface heterogeneity of the conductivity field control the HSHM statistics, for example, the time expected for the probability of the HSHM to occur to reach a-priori set thresholds or time to peak probability.

        Lastly, as mentioned both here and in previous studies, statistical methods for quantifying the occurrences of

HSHMs and the associated uncertainties are needed to advance our understanding of the mechanisms that cause HSHMs, as well as to enhance our ability to predict HSHMs and manage their consequences.

**Acknowledgments**

        We gratefully acknowledge the Jane Lewis Fellowship Committee of the University of California, Berkeley, and the Earth and Environmental Science Division at Lawrence Berkeley National Laboratory for their generous

support through fellowships awarded to the first author. The second author's work is supported by the Office of Science, Office of Advanced Scientific Computing, as part of the project "Deduce: Distributed Dynamic Data Analytics Infrastructure for Collaborative Environments" under contract no. DE-AC02-05CH11231. The third author acknowledges funding from the Italian Ministry of Education, University and Research (MIUR) in the frame of the Departments of Excellence Initiative 2018—2022 granted to DICAM of the University of Trento. As this study

focused on theory development, data were not used, nor created for this research. All the information needed for evaluating and checking the results is provided in the paper.

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

**Appendix**

**A1. Equations for the displacement pdf**

Assuming steady, uniform in the average flow with mild heterogeneity of the log-hydraulic conductivity field with Gaussian displacement, the displacement pdf in the longitudinal direction $(x_1)$ is given by the following equation (Dagan and Nguyen, 1989; Dagan and Rubin, 1992):

$$f_{X_1(t^*)}(x_1) = \frac{1}{\sqrt{2\pi X_{11}(t^*)}} \exp\left[-\frac{1}{2}\frac{(x_1 - Ut^*)^2}{X_{11}(t^*)}\right]. \tag{A1}$$

Additionally, the displacement pdf in the transverse directions $(x_2$ and $x_3)$ is given by

$$f_{X_i(t^*)}(x_i) = \frac{1}{\sqrt{2\pi X_{ii}(t^*)}} \exp\left[-\frac{1}{2}\frac{x_i^2}{X_{ii}(t^*)}\right], i = 2,3. \tag{A2}$$

**A2. Equations for displacement variances under anisotropic conditions**

Dagan (1984) developed a solution for the displacement variances for an exponential and axisymmetric covariance function:

$$C_Y(\boldsymbol{r}) = \langle(Y(\boldsymbol{x}) - \langle Y\rangle)(Y(\boldsymbol{x} + \boldsymbol{r}) - \langle Y\rangle)\rangle = \exp\left[-\sqrt{\frac{r_1^2 + r_2^2}{I_{Yh}^2} + \frac{r_3^2}{I_{Yv}^2}}\right], \tag{A3}$$

$$X_{11} = \sigma_Y^2 I_Y^2 \{2t^* + 2[\exp(-t^*) - 1] + 8e\int_0^\infty [\bar{J}_0(Kt^*) - 1]$$

$$\cdot\left[\frac{1}{(1 + K^2 - e^2 K^2)^2} - \frac{eK}{(1 + K^2 - e^2 K^2)^2(1 + K^2)^{0.5}} - \frac{eK}{2(1 + K^2 - e^2 K^2)(1 + K^2)^{1.5}}\right]dK$$

$$-2e\int_0^\infty \left[\bar{J}_0(Kt^*) - \frac{\bar{J}_1(Kt^*)}{Kt^*} - \frac{1}{2}\right]\cdot\left[\frac{e^3 K^3(e^2 K^2 - 5 - 5K^2)}{(e^2 K^2 - 1 - K^2)^3(1 + K^2)^{1.5}} + \frac{1 + K^2 - 5e^2 K^2}{(1 + K^2 - e^2 K^2)^3}\right]dK\}, \tag{A4}$$





$$X_{22} = -2e\sigma_Y^2 I_Y^2$$

$$\cdot \int_0^\infty \left[\frac{\overline{J_1}(Kt^*)}{t^*} - \frac{K}{2}\right]\left[\frac{e^3 K^2(e^2 K^2 - 5K^2 - 5)}{(e^2 K^2 - 1 - K^2)^3(1 + K^2)^{1.5}} + \frac{1 + K^2 - 5e^2 K^2}{K(1 + K^2 - e^2 K^2)}\right]dK, \qquad (A5)$$

$$X_{33} = -4e\sigma_Y^2 I_Y^2 \int_0^\infty [\overline{J_0}(Kt^*) - 1]$$

780

$$\cdot \left\{\frac{1}{(e^2 K^2 - 1 - K^2)^2}\left[\frac{1}{2} + \frac{2e^2 K^2}{1 + K^2 - e^2 K^2} + \frac{eK(e^2 K^2 + 3 + 3K^2)}{2(e^2 K^2 - 1 - K^2)(1 + K^2)^{0.5}}\right]\right\}dK. \qquad (A6)$$

where $r$ is the two-point separation distance and $\langle Y \rangle$ the ensemble mean of the log-conductivity $Y = \ln K$. $\overline{J_0}$ and $\overline{J_1}$ are, respectively, the zero and first order of the first kind Bessel functions.

### A3. Equations for displacement variances under isotropic conditions

Dagan (1984) provided analytical solutions for longitude and transverse displacement variances. This is a
785    special case for the anisotropic case with $e = 1$.The solutions are as follows:

$$X_{11} = \sigma_Y^2 I_Y^2 \left\{2t^* - 2\cdot\left[\frac{8}{3} - \frac{4}{t^*} + \frac{8}{t^{*3}} - \frac{8}{t^{*2}}\left(1 + \frac{1}{t^*}\right)\exp(-t^*)\right]\right\}. \qquad (A7)$$

$$X_{22} = X_{33} = 2\sigma_Y^2 I_Y^2 \left[\frac{1}{3} - \frac{1}{t^*} + \frac{4}{t^{*3}} - \left(\frac{4}{t^{*3}} + \frac{4}{t^{*2}} + \frac{1}{t^*}\right)\exp(-t^*)\right]. \qquad (A8)$$