# Peer review of "Statistical Characterization of Environmental Hot Spots and Hot Moments and Applications in Groundwater Hydrology"

_Hydrology and Earth System Sciences, 2020_

## Referee Comment (RC1) · Anonymous Referee #1 · 30 Dec 2020

This paper, "Statistical Characterization of Environmental Hot Spots and Hot Moments and Applications in Groundwater Hydrology" by Chen and others proposes statistical methods for characterizing ecosystem control points primarily in an ecohydrological context. The paper has many interesting ideas and promising approaches. However, I found the current manuscript structure to be quite convoluted and difficult to follow. With a substantial restructuring around a clear set of hypotheses, I think this paper would be highly appropriate for this journal. I provide some general comments and suggestions below.

1. The paper flips back and forth multiple times among general discussion, case stud-

[Figure]

ies, and equations. While this could work, the manuscript needs a clear through line and arc to pull this off. The current sections seem jumbled, and I found myself unsure of what was novel versus what was being reviewed. This is particularly notable at the conclusion of the paper, where general comments are made that are very similar to the introduction (i.e., the paper does not build on itself or at least not in a way that invites readers to come along). I would recommend a careful restructuring with clear delineations between the new content and previous work.

2. In the introduction, the authors provide a useful summary of the use of the hot spot and hot moment terminology in several fields, including medicine. The authors claim to combine these definitions around line 40: "we combined these definitions such that, henceforth, HSHMs are referred to as rare locations or events that could exert a disproportionate influence on an ecosystem and which are associated with heightened health or environmental risks." This definition is too broad to be useful, in my opinion. For example, what does disproportionate mean in a quantitative sense? Both the original HSHMs paper (McClain et al., 2003) and the recent conceptual update (Bernhardt et al., 2017) emphasize the idea of variable influence, characterized as response surfaces or distributions. Could the authors of this study useful formalize this definition based on a certain percentage or proportion relative to a process and spatiotemporal domain of interest? For example, is a site that has a denitrification rate that is in the 60th percentile of the studied ecosystem a control point? Can the "hotness" of an ecological process be characterized by its spatiotemporal lumpiness in an easily interpretable way?

3. One of the justifying premises of the paper is that there are not available statistical tools to describe HSHM phenomena (e.g. lines 91 through 93). I think this is generally a weak justification and in this case, clearly incorrect. There are many theoretical and empirical models to simulate HSHM dynamics, including several in just the past few years (Druhan et al., 2014; Abbott et al., 2016; Kolbe et al., 2019; Li et al., 2017; Oldham et al., 2013; Pinay et al., 2015; Zarnetske et al., 2012; Bochet et al., 2020).

Better integrating this work with the existing literature would clarify what is novel about the contribution and help the reader get up to speed on this important topic.

References Abbott, B.W., Baranov, V., Mendoza-Lera, C., Nikolakopoulou, M., Harjung, A., Kolbe, T., Balasubramanian, M.N., Vaessen, T.N., Ciocca, F., Campeau, A., Wallin, M.B., Romeijn, P., Antonelli, M., Gonçalves, J., Datry, T., Laverman, A.M., de Dreuzy, J.-R., Hannah, D.M., Krause, S., Oldham, C., Pinay, G., 2016. Using multi-tracer inference to move beyond single-catchment ecohydrology. Earth-Sci. Rev. 160, 19–42. https://doi.org/10.1016/j.earscirev.2016.06.014 Bernhardt, E.S., Blaszczak, J.R., Ficken, C.D., Fork, M.L., Kaiser, K.E., Seybold, E.C., 2017. Control Points in Ecosystems: Moving Beyond the Hot Spot Hot Moment Concept. Ecosystems 20, 665–682. https://doi.org/10.1007/s10021-016-0103-y Bochet, O., Bethencourt, L., Dufresne, A., Farasin, J., Pédrot, M., Labasque, T., Chatton, E., Lavenant, N., Petton, C., Abbott, B.W., Aquilina, L., Borgne, T.L., 2020. Iron-oxidizer hotspots formed by intermittent oxic–anoxic fluid mixing in fractured rocks. Nat. Geosci. 1–7. https://doi.org/10.1038/s41561-019-0509-1 Druhan, J.L., Steefel, C.I., Conrad, M.E., DePaolo, D.J., 2014. A large column analog experiment of stable isotope variations during reactive transport: I. A comprehensive model of sulfur cycling and $\delta$34S fractionation. Geochim. Cosmochim. Acta 124, 366–393. https://doi.org/10.1016/j.gca.2013.08.037 Kolbe, T., Dreuzy, J.-R. de, Abbott, B.W., Aquilina, L., Babey, T., Green, C.T., Fleckenstein, J.H., Labasque, T., Laverman, A.M., Marçais, J., Peiffer, S., Thomas, Z., Pinay, G., 2019. Stratification of reactivity determines nitrate removal in groundwater. Proc. Natl. Acad. Sci. 201816892. https://doi.org/10.1073/pnas.1816892116 Li, L., Maher, K., Navarre-Sitchler, A., Druhan, J., Meile, C., Lawrence, C., Moore, J., Perdrial, J., Sullivan, P., Thompson, A., Jin, L., Bolton, E.W., Brantley, S.L., Dietrich, W.E., Mayer, K.U., Steefel, C.I., Valocchi, A., Zachara, J., Kocar, B., Mcintosh, J., Tutolo, B.M., Kumar, M., Sonnenthal, E., Bao, C., Beisman, J., 2017. Expanding the role of reactive transport models in critical zone processes. Earth-Sci. Rev. 165, 280–301. https://doi.org/10.1016/j.earscirev.2016.09.001 McClain, M.E., Boyer, E.W., Dent, C.L.,

[Figure]

Gergel, S.E., Grimm, N.B., Groffman, P.M., Hart, S.C., Harvey, J.W., Johnston, C.A., Mayorga, E., McDowell, W.H., Pinay, G., 2003. Biogeochemical Hot Spots and Hot Moments at the Interface of Terrestrial and Aquatic Ecosystems. Ecosystems 6, 301–312. https://doi.org/10.1007/s10021-003-0161-9 Oldham, C.E., Farrow, D.E., Peiffer, S., 2013. A generalized Damköhler number for classifying material processing in hydrological systems. Hydrol. Earth Syst. Sci. 17, 1133–1148. https://doi.org/10.5194/hess-17-1133-2013 Pinay, G., Peiffer, S., De Dreuzy, J.-R., Krause, S., Hannah, D.M., Fleckenstein, J.H., Sebilo, M., Bishop, K., Hubert-Moy, L., 2015. Upscaling Nitrogen Removal Capacity from Local Hotspots to Low Stream Orders' Drainage Basins. Ecosystems 18, 1101–1120. https://doi.org/10.1007/s10021-015-9878-5 Zarnetske, J.P., Haggerty, R., Wondzell, S.M., Bokil, V.A., González-Pinzón, R., 2012. Coupled transport and reaction kinetics control the nitrate source-sink function of hyporheic zones. Water Resour. Res. 48, W11508. https://doi.org/10.1029/2012WR011894

---

## Referee Comment (RC2) · Anonymous Referee #2 · 27 Feb 2021

This study focuses on a very important and difficult topic (i.e., hot spots and hot moments: HSHMs). All progress in this domain is appreciated and important. They develop a statistical approach for characterizing HSHMs and work through some demonstration cases. As they state, "...clear statistical conventions of HSHMs are missing, which significantly limits the transferability of these approaches." I agree, and the topic of this contribution is relevant to a broad range of researchers and other practitioners due to the generality of the framework. After reading the manuscript I am left conflicted, however. While I appreciate the topic, I don't have a strong grasp of why this framework is needed. In some cases the framework seems based on straightforward 'if else' conditions that depend on already knowing what governs HSHMs. I also worry about the

focus on defining places in space and time as either being or not being a HSHM. Rates are binary like that, and I am worried we are making arbitrary cutoffs of what counts as a HSHM instead of embracing the more continuous nature of natural systems. The proposed framework in based squarely on a binary view of HSHMs, which to me is limiting. There were many times in the manuscript that I did not understand how this framework would be used to advance understanding, and after reading the whole thing I am still mostly unclear. There are a couple examples in the end showing how it could be used, but I think those examples could by sampling spatial and temporal dynamics of physically based simulations. I don't understand why the statistical framework is needed or how to put into practice. I am willing to believe that I am just missing something and am willing to be convinced. I would highly recommend the authors make the applications more real world relevant. Speak to the average field person that is interested in HSHMs and tell them how they can use this framework and why it is better than other approaches. I think without plain language connections to the center of mass of researchers, the framework will go mostly unused. Below I provide more comments, many of them related to the above summary points.

Line 40: I am having a hard time with the definition used for HSHMs as it depends on the event having a negative effect on something (health or environment). But what if a HSHM does something beneficial like remove a pollutant. Is that not considered here? Maybe a slight modification of the definition is all that's needed.

Line 90: Here again the focus is on the negative side of HSHMs, I suggest taking a more balanced view that includes their benefits as well.

Line 100: I would suggest adding 1 sentence providing a non-jargon definition of indicator statistics. It will make the work more accessible.

Line 135: While I appreciate the development of a rigorous statistical framework, I question the utility of the binary definition of whether a place in space/time is or is not a HSHM. Do we care more about the definition or its influence? The influence

is some continuous function of the magnitude to which it deviates from background conditions. It would be more powerful to define a statistical framework that captured this more continuous perspective. At a minimum, I think the authors should discuss this limitation of their framework and provide ideas for developing a more continuous approach. For example, maybe one could continuously vary the Cth and Rth from equation 1 to examine outcomes across a continuum of thresholds?

Line 145-150: All examples are for concentrations (Cth). It would be good to provide some examples for rates (Rth).

Line 160: Please briefly explain why type B includes the spatial component instead of only including the temporal.

Equation 3: This seems a big circular to me. It seems like this says that a location is a hot spot because it has the conditions (e.g., concentration) needed to be a hot spot given our defined threshold of what counts as a hot spot. So it's a hot spot because it's a hot spot. Maybe this can be clarified in terms of how this isn't circular? In other words, explain further why it is useful to call some place a hot spot based on defined criteria. Why don't we just define the location based on it's levels of continuous variables relevant to a given situation? This goes back to my comment above about the very binary nature of this approach. I am not yet convinced that this is really moving us forward a great deal. Though I am keeping an open mind as I read.

Line 225: I like the statistical framework here, though it presumes that we have complete (or very good) knowledge of the spatial and temporal factors governing HSHM 'activation' and I wonder if that makes this framework difficult to use? That is, if we already know the conditions that lead to HSHM behavior, then we already know that, and I am not clear on what we are learning from this framework.

Line 265: Again I am not understanding what we are learning here. The hot spots have been defined as NRZ with specific quantitative conditions. So what more is equation 10 telling us? I was expecting to see a figure or analysis here that went to the next

level of understanding through the use of eq 10.

Line 340: Unless I am missing something, the examples are based around meeting specific conditions in space and/or time, and saying that if all conditions are met, then a HSHM should occur. That's all fine, but again, what are we learning from that? It seems like this boils down to an if-else statement that is built around previous analyses of a given system. That seems really straightforward to the point that I feel like I am missing something. Maybe more of the implications can be drawn out through these sections?

Section 4.2.3: To be honest, I am not savvy enough to follow the math in this section. I can only assume that is it correct, and maybe other reviewers can go through it. Regardless of whether it is correct or not, however, I do not understand the purpose of the formulations. Maybe they come clear further into the paper. At this point this section and the previous two seem esoteric, and I am not sure what the work is really driving towards.

Section 4.3: Not clear on what 'w' is in this case. More generally, I continue to struggle to understand what we are learning. I really want to get on board and I feel strongly about HSHMs as important features, I just am struggling to connect the conceptual dots.

Line 515: can you show a figure of this? pretty hard to understand as is.

Line 520-540: Okay, so now we start to see some results from the framework, in which the time course of HSHM development is linked to variation in conductivity. This is nice, though I must wonder whether the formal statistical framework is necessary. Could this be done just as well with a Monte Carlo approach? What I am missing is a convincing argument that the formal framework is needed. Could one not just run a simulation and sample it to characterize the spatial distribution of biogeochemical rates, use that to determine the frequency and magnitude of hot spots and then do that through time to show the time course?

Line 595: I don't recall seeing any results showing how the framework can be used to study uncertainty. This seems important, but not presented.

Line 605: I think it would be useful to expand on the discussion through the manuscript in terms of how the framework provides understanding of mechanisms. Through much the paper it seemed that the mechanisms were known a priori and were actually used to define conditions that result in HSHMs. I don't fully understand how we are gaining more mechanistic understanding, but I am open to hearing more.

---

## Author Comment (AC1) · 19 Mar 2021

*Interactive comment* on "Statistical Characterization of Environmental Hot Spots and Hot Moments and Applications in Groundwater Hydrology" by Jiancong Chen et al.

Jiancong Chen et al.
nigel_chen1993@berkeley.edu
Corresponding to rubin@ce.berkeley.edu

We thank the reviewer for reviewing our manuscript and appreciate all the helpful suggestions and comments. Below, we present our responses to reviewer's comments and the plan to revise the manuscript correspondingly.

1. **Restructuring the paper**. In the revised manuscript, we will improve the readability by streamlining the introduction and the objectives of the work, followed by a much clearer separation between the presentation of the framework, the illustrative examples and the discussion, as requested by the reviewer. Any duplication will be carefully handled in the revised manuscript.

2. **Definition of 'disproportionate'**. Based on the complex nature of HSHMs, it is not practical and beneficial to look for a universal mathematical definition using percentage or proportion. For example, a $60^{th}$ percentile denitrification rate may trigger HSHM at a riparian site, but may fail to indicate HSHM condition at other sites as other static and/or dynamic factors can also control HSHM occurrences significantly. However, our proposed statistical framework provides a very flexible approach that can incorporate different types of HSHMs through static and dynamic contributors, modeled with indicator random variables and stochastic processes. For example, the permanent control points (as defined by Bernhardt et al., 2017) can be modeled with static-only indicators whereas activated control points will require both static and dynamics indicators. The cutoff values, percentage or proportion are defined by users and can be modified at will, as in equation (1). One can change these quantities based on prior information, risk tolerance or through statistical quantities. For certain HSHMs that have negative influences on ecosystems, thresholds are often introduced in environmental regulations in order to identify levels of contamination above which to consider a site as contaminated. In addition, activation thresholds may be used for chemical reactions that are necessary for biogeochemically driven HSHMs.

3. **Novelty**. We appreciate the reviewer's recommendations of relevant references and studies. We agree with the reviewer that there are many theoretical, empirical models and experimental approaches dealing with the HSHM dynamics. There are also various approaches to define 'hotness' as summarized in Bernhardt et al. (2017), such as simple comparison to average or matrix; substantial percentage of total flux; outlier in distribution of data; statistically significant difference between or among landscape elements or time periods categorized a priori; and contribution to flux/total area or time. However, most of these quantitative methods are derived based on site-specific data or simulation results, which limits the transferability from one site to other sites; and from one type of HSHM to other types of HSHMs. Thus the challenge is not how we define a single cutoff value for a specific HSHM at a specific site, but rather to develop a

statistical framework, capable of handling a generality of cases, and therefore progressing beyond local conditions. From this perspective, our proposed framework is novel and beneficial for future HSHM studies, summarized as follows: (a) With the indicator formulation, the framework is flexible enough to handle different scenarios of cutoff values (see point #2); (b) Our proposed framework is unified and allows us to investigate HSHMs under conditions of uncertainty; (c) Our framework can integrate results from HSHM studies using different approaches, whether results from Monte-Carlo simulations or direct data based quantifications; (d) Probabilities are assigned to the entire domain and time of HSHM concerns and modeled with corresponding stochastic processes; (e) Our framework can be easily integrated with Bayesian concepts such as conditioning as well as utilization of prior information from other sites. Based on the above points, we believe the statistical framework can make contributions to the HSHM community. In the revised manuscript, we will clearly outline these advantages.

---

## Author Comment (AC2) · 19 Mar 2021

*Interactive comment* on "Statistical Characterization of Environmental Hot Spots and Hot Moments and Applications in Groundwater Hydrology" by Jiancong Chen et al.

Jiancong Chen et al.
nigel_chen1993@berkeley.edu
Corresponding to: rubin@ce.berkeley.edu

We appreciate the reviewer's efforts in reading our manuscript and providing useful comments and recommendations. We will provide summary notes to the reviewer's general comments and responses to the reviewer's detailed comments and questions. Please see below.

Summary notes:
1. **Why stochastic?** The processes governing the HSHMs are very likely subject to some uncertainty. There may be uncertainty in the parameters and also in the governing equations. This is the reason to adopt a stochastic formulation. In this way, the uncertainty can be modeled and even reduced by taking advantage of the information provided by the data. The mechanism for modeling and reducing uncertainty are built into our approach. For example, we can use prior information from similar sites, and we can use local measurements for conditioning.

2. **Defining places in space and time as either being or not being HSHMs**. Our characterization of HSHMs is not binary, because we use probabilities. In our approach, all space and time intervals in the investigated domain are associated with probabilities to be or not to be a HSHM. Notice that the location of the HSs is uncertain due to the combined effect of physical system heterogeneity and limitations in its characterization. In addition, even if the positions of the HSs are known without uncertainty, hot moments may depend on other factors (e.g., solute pathway, retention time), which can also be uncertain.

3. **Arbitrary cutoffs**. In our approach, the cutoffs are defined by the user and can be modified at will. One can change the cutoff values based on prior information and based on risk tolerance. For HSHMs that have negative influences, thresholds are often introduced in environmental regulations in order to identify levels of contamination above which to consider a site as contaminated. In addition, activation thresholds may be used to identify the thresholds for reactions that are necessary for biogeochemical driven HSHMs (see also point 5 below).

4. **Binary view of HSHMs.** As stated in point 2, we model the HSHMs stochastically. For example, we can have a zone with high probability next to other zones with lower probabilities in terms of HSHMs occurrence. Thus, we do adopt a continuum approach by creating HSHM probability maps. In another note, we suggest that there might be situations that require focusing on a particular area because of a need to focus on the site investigation efforts. Thus, in our approach, we can identify areas that are more critical/sensitive compared to others, and this could assist the project managers in defining priorities. For example, at the Rifle site (Wainwright et al. 2010), geophysical

datasets indicated the presence of naturally reducing zones (NRZs), which may have higher level of uranium and nitrate. Based on this information, site investigation and parameter estimation were both goal oriented, which reduces efforts and uncertainties in quantifying the corresponding HSHMs.

5. **Improving understanding**. We will expand our discussion in order to improve understanding. In particular, we make the following points: (a) Our framework can be used to investigate HSHM sites and identify the process and parameters controlling the HSHMs. (b) As the reviewer noted, there's uncertainty associated with the HSHMs. Using our approach, we can identify which models and parameters work, using for example Bayesian model comparison and identifying the best performing models or whether the current understanding of a certain HSHM is lacking. (c) The probabilistic approach offers great advantages of addressing the uncertainty on HSHMs and reducing it. (d) We will also add information on where to get the threshold parameters.

6. **Why a statistical framework**? Based on the experiences in the hydrology community, the coupling of probabilistic concepts with the physics led to a tremendous progress in our ability to model the complex phenomena taking place in the subsurface. Similar observations have been made on multiple disciplines within earth sciences. There is a vast body of knowledge accumulated in hydrology and what we want to show in this paper is that this knowledge could also bring enormous potential to HSHMs investigations.

7. **Simple language**. We will add plain language discussion in the revised manuscript.

In the following section, we will provide a detailed response to the reviewer's specific comments and questions.

Detailed responses:
*L40: I am having a hard time with the definition used for HSHM as it depends on the event having a negative effect on something (health or environment). But what if a HSHM does something beneficial like remove a pollutant. Is that not considered here? Maybe a slight modification of the definition is all that's needed.*

Response: We will modify the definition to include the beneficial perspectives (L41). Table 1 in the revised paper will exhibit an extra column indicating whether a specific HSHM has a positive, neutral or negative impact on the ecosystem.

*L90: Here again the focus is on the negative side of HSHMs, I suggest taking a more balanced view that includes their benefits as well.*

Response: We will expand the HSHM definition to cover both positive and negative perspectives of HSHMs. Please also see our response to L40 comment.

*L100: I would suggest adding 1 sentence providing a non-jargon definition of indicator statistics. It will make the work more accessible.*

Response: We agree with the reviewer and will make necessary modifications to make the reading more accessible.

*L135: While I appreciate the development of a rigorous statistical framework, I question the utility of the binary definition of whether a place/time is or is not a HSHM. Do we care more about the definition or its influence? The influence is some continuous function of the magnitude to which it deviates from background conditions. It would be more powerful to define a statistical framework that captured this more continuous perspective. At a minimum, I think the authors should discuss the limitation of their framework and provide ideas for developing a more continuous approach. For example, maybe one could continuously vary the Cth and Rth from equation 1 to examine outcomes across a continuum of thresholds?*

Response: As discussed in items 2, 3 and 4 above, our framework is flexible as it can incorporate different conditions that trigger HSHMs. The cutoff values are chosen by users and can be modified at will. With this flexibility, one could definitely vary $C_{th}$ and $R_{th}$ values, and examine how the probability of HSHM occurrences changes correspondingly, as suggested by the Reviewer. Thus our approach indeed captures the continuous perspectives as specified in the previous answers.

*L145-L150: All examples are for concentrations (Cth). It would be good to provide some examples for rates (Rth).*

Response: In the revised manuscript, we will include multiple examples for rates. For example, $R_{th} = 0$ can be used for chemical reactions that have significant negative impact on ecosystem (e.g., nuclear reactions). $R_{th}$ values can also be obtained based on similar studies, such as denitrification and carbon cycling rates summarized in Harms and Grimm (2008).

*L160: Please briefly explain why type B includes the spatial component instead of only including the temporal.*

Response: The idea here is to identify where the dynamic conditions exist in conjunction with spatial zones to trigger a hot spot. For example, following the reviewer's comments, a zone of high concentration may be a location for HSHMs, only that we do not know where it gets triggered. An example here could be nuclear waste remediation sites where natural attenuation strategies are in place. While contaminants can be held in place; within the zones where contamination occurred, yet some temporal conditions may trigger the formation of these HSHMs. Here it is worthwhile to note both the temporal conditions, but also the spatial domain of HS. Furthermore, the hot spot may also depend on variables different from the concentration of the species of original interest. For example, a nuclear contamination site that has historically looked at uranium can now be potential hot spots for strontium.

*Equation 3: This seems a bit circular to me. It seems like this says that a location is a hot spot because it has the conditions (e.g., concentration) needed to be a hot spot given our*

*defined threshold of what counts as a hot spot. So it's a hot spot because it's a hot spot. Maybe this can be clarified in terms of how this isn't circular? In other words, explain further why it is useful to call some place a hot spot based on defined criteria. Why don't we just define the location based on its levels of continuous variables relevant to a given situation? This goes back to my comment above about the very binary nature of this approach. I am not yet convinced that this is really moving us forward a great deal. Though I am keeping an open mind as I read.*

Response: As mentioned in L160 response, the location in time and space may be unknown. Equation (3) is related to spatial variability and uncertainty in site characterization, which leads to uncertainty in identifying the locations critical for HSHMs. The definition is needed to define the corresponding statistical random variables.

*L225: I like the statistical framework here, though it presumes that we have complete (or very good) knowledge of the spatial and temporal factors governing HSHM 'activation' and I wonder if that makes this framework difficult to use? That is, if we already know the conditions that lead to HSHM behavior, then we already know that, and I am not clear on what we are learning from this framework.*

Response: We agree with the reviewer that work in the HSHM community thus far has been remarkably site-specific. To enable the transferability of HSHM features from one site to the other, we have proposed this statistical framework. For example, static indicators at riparian sites could be quite similar – riparian buffer strips or microtopographic depressions. By using a statistical formulation to capture these spatial zones and applying them to a new site under corresponding dynamic conditions can help us pre-identify potential zones of HSHMs. It is also important to note that the impact of HSHMs does not depend only on the fact that they may exist in a certain compartment, i.e. riparian and hyporheic zones, but also on their location and duration in the active state, which may be intermittent. All these factors are uncertain because we don't know the exact location of HS and for how long they are active under new conditions and new sites. Thus, our stochastic approach is beneficial to enhance applicability to other sites.

*L265: Again I am not understanding what we are learning here. The hot spots have been defined as NRZ with specific quantitative conditions. So what more is equation 10 telling us? I was expecting to see a figure or analysis here that went to the next level of understanding through the use of eq. 10.*

Response: Equation 10 is an example how we can construct a static indicator quantitatively. Please also see our responses to L225 comment.

*L340: Unless I am missing something, the examples are based around meeting specific conditions in space and/or time, and saying that if all conditions are met, then a HSHM should occur. That's all fine, but again, what are learning from that? It seems like this boils down to an if-else statement that is built around previous analyses of a given*

*system. That seems really straightforward to the point that I fell like I am missing something. Maybe more of the implications can be drawn out through these sections?*

Response: There is a challenge in knowing what the conditions required to trigger HSHMs are, but knowing the conditions may not suffice to predict when and where. This is where our proposed approach comes in. As mentioned above, our proposed framework is unified and allows us to investigate a variety of HSHMs, with complex dynamics multi-dimensional dynamics and under diverse conditions of uncertainty. It also can be easily integrated with the concept of Bayesian conditioning in order to reduce uncertainty and to develop site investigation schemes with information from similar sites. We do not promulgate an if-else approach, because we assign probabilities over the entire domain. The proposed equations and formula in this section are mainly presented to show how the framework can be utilized and how the corresponding indicators can be constructed.

*Section 4.2.3: To be honest, I am not savvy enough to follow the math in this section. I can only assume that it is correct, and maybe other reviewers can go through it. Regardless of whether it is correct or not, however, I do not understand the purpose of the formulations. Maybe they come clear further into the paper. At this point this section and the previous two seem esoteric, and I am not sure what the work is really driving towards.*

Response: In previous sections we focused on evaluating the probability of HSHMs occurring at a given time $t$. This allows us to evaluate when and where we could observe the highest probability of HSHM occur. As hot moments can persist over time periods, estimating the corresponding probabilities for given time intervals becomes also quite important. And this is the main reason for introduced section 4.2.3.

Specifically, Equation (22) – (26) describes the dynamic indicator and an analytical stochastic solution for the HSHM. These equations can be simplified into Equation (27) – (30) if the hot spot can be defined by a simple geometry as described in Line 458. The deviations of these equations are based on stochastic theories, which are well documented and extensively verified cf., Dagan. (1989) and Rubin. (2003).

*Section 4.3: Not clear on what 'w' is in this case. More generally, I continue to struggle to understand what we are learning. I really want to get on board and I feel strongly about HSHMs as important features, I just am struggling to connect the conceptual dots.*

Response: In most world conditions, HSHMs occur within a volume rather than a single spot. And this is why we introduced 'w' in the mathematical formulations, which represents the corresponding dimensions of this control volume.

*L515: Can you show a figure of this? pretty hard to understand as is.*

Response: Hot spot $\Omega$ was placed $21I_{YH}$ away from the source, and the dimension of $\Omega$ is $(2I_{YH}, 2I_{YH}, 2I_{YV})$. Figure 3 presents the configuration of this example, where the red box is the candidate hot spot $\Omega$.

*L520-540: Okay, so now we start to see some results from the framework, in which the time course of HSHM development is linked to variation in conductivity. This is nice, though I must wonder whether the formal statistical framework is necessary. Could this be done just as well with a Monte Carlo approach? What I am missing is a convincing argument that the formal framework is needed. Could one not just run a simulation and sample it to characterize the spatial distribution of biogeochemical rates, use that to determine the frequency and magnitude of hot spots and then do that through time to show the time course?*

Response: Our approach can definitely be applied using Monte-Carlo (MC) approaches. We present a framework, and it can be applied using analytical models (when available) or using MC simulation. These approaches are complementary rather than exclusive. One can use our framework to define the flowchart for the Monte-Carlo analysis. Although MC approach can be used for implementation, our approach goes further than MC because it can easily incorporate Bayesian concepts such as conditioning as well as utilization of prior information from other sites. For example, knowledge from previous nitrogen HSHM studies can be implemented with the proposed framework and guide new HSHM investigation at other new sites. More details will be provided in the revised manuscript.

*L595: I don't recall seeing any results showing how the framework can be used to study uncertainty. This seems important, but not presented.*

Response: As we stated in point 1 and 6 above, our proposed framework incorporate uncertainty through modeling the dynamics as stochastic processes and through modeling the parameters as random variables. For example, in Section 4.4, we show how the uncertainty surrounding the hydraulic conductivity influences the probability of HSHM occurrence in the subsurface.

*L605: I think it would be useful to expand on the discussion through the manuscript in terms of how the framework provides understanding of mechanisms. Through much the paper it seemed that the mechanisms were known a priori and were actually used to define conditions that result in HSHMs. I don't fully understand how we are gaining more mechanistic understanding, but I am open to hearing more.*

Response: We appreciate this comment. In the revised manuscript, we have incorporated certain Bayesian statistics theories into the indicator formulation and expanded the discussion correspondingly. The wide range of approaches used for modeling HSHMs reported in the literature are helpful in gaining a better understanding of HSHMs, however it is challenging to evaluate and rank the suitability of the models for realistic scenarios. This is where our study becomes useful. The flexibility of our proposed framework enables us to compare the performance of competing models and select appropriate models for new sites. And we would cite here a couple of examples. First, Bayesian model averaging approaches (Volinsky et al., 1999) could be implemented to obtain a combined and less risky estimation of HSHMs at new sites. Second, model

comparison criteria, such as the Akaike information criteria (AIC, Akaike, 1974) and Bayesian information criteria (Schwarz, 1978) can also be applied to compare and rank the performance of different HSHM indicator models and their ability to explain observations. For example, smaller AIC and BIC values indicate a better match between a HSHM model and data. Large AIC and BIC values would suggest an incomplete and possibly even faulty model. Through this process of model inter-comparison, we could gain better understanding of the underlying mechanism, which, in essence, is the learning process that the reviewer rightfully wishes us to show.

Reference:
Akaike, H.: A New Look at the Statistical Model Identification, IEEE Trans. Automat. Contr., doi:10.1109/TAC.1974.1100705, 1974.
Dagan, G.: Flow and Transport in Porous Formations., Springer Verlag, Berlin., 1989.
Harms, T. K. and Grimm, N. B.: Hot spots and hot moments of carbon and nitrogen dynamics in a semiarid riparian zone, J. Geophys. Res. Biogeosciences, 113(1), 1–14, doi:10.1029/2007JG000588, 2008.
Rubin, Y.: Applied Stochastic Hydrogeology, Oxford University Press, Oxford, UK., 2003.
Schwarz, G.: Estimating the Dimension of a Model, Ann. Stat., doi:10.1214/aos/1176344136, 1978.
Volinsky, C. T., Raftery, A. E., Madigan, D. and Hoeting, J. A.: David Draper and E. I. George, and a rejoinder by the authors, Stat. Sci., 14(4), 382–417, doi:10.1214/ss/1009212519, 1999.

---

## Referee Report (RR1)

HESS-2020-343

Statistical Characterization of Environmental Hot Spots and Hot Moments and Applications in Groundwater Hydrology

Jiancong Chen, Bhavna Arora, Alberto Bellin, and Yoram Rubin

The paper presents present a probabilistic formulation of hot spots and hot moments (HSHMs) geared towards hydrology and in particular, though not exclusively, groundwater applications. There is a substantial literature on this topic in environmental engineering, yet upon reading what is a complete introduction, the reader gets the impression that the approach proposed by the authors is substantially more general than most previous applications. This impression is substantiated by the methodology presented in the following, which views HSHMs occurrence as a binary event in space/time and identifies indicator geostatistics as the tool of choice to adopt. A Bernoulli distribution is chosen to model the binary random variable, and this is due to a specific property of the Bernoulli distribution, which somewhat detracts from the generality of the model. A positive aspect of the methodology presented is the option to adopt different choices for activation thresholds: concentration, reactivity, mass, flux, percentiles, extremes. Three different HSHM categories are identified, i.e. induced by a) static, b) static and dynamic indicators, and c) multiple dynamic indicators. The stochastic formalism allows comparing alternative HSHM via information or Bayesian information criteria. Several examples are described via specific case studies for the three different categories and for groundwater hydrology. A more detailed illustrative example to groundwater hydrology based on classical stochastic hydrology theory valid under ergodicity is then illustrated. Wider pdfs of the indicator are associated with larger logconductivity variances, and a convincing explanation is presented in detail. The Appendix presents a recapitulation and/or extension of classical results in stochastic subsurface hydrology.

The paper looks as a mature contribution almost ready for publication. Given the topic and the type of paper, I see little room for further improvement. Results are of interest to the readership of *Hydrology and Earth System Sciences*. The methods are adequate, the paper subdivision into sections sound, and the figures illustrative. I reviewed only the revised version, but I can tell there were substantial improvements and clarifications upon looking at the earlier version and to the extent of the modifications. On the reviewers side, the remarks were extensive and provided a noteworthy input to the paper quality.

There is a general question that deserves the author's response, i.e.:

- How crucial is the assumption of a Bernoulli distribution for the indicator variable ? Could they develop a more general theory without it, maybe subject to other limitations ?

and a minor correction:

- Note that at line 425, 4.3 → 4.2.

---

## Author Response (AR2)

We thank the editor and two anonymous referees for reviewing our manuscript and providing us with constructive feedback. Below are our point-by-point responses to the referees' comments, followed by a revised manuscript showing all track changes made through the minor revision. Note referees' original comments are in italic and all line numbers are based on the revised manuscript.

Response to RC#1

*Summary: The paper presents a probabilistic formulation of hot spots and hot moments (HSHMs) geared towards hydrology and in particular, though not exclusively, groundwater applications. There is a substantial literature on this topic in environmental engineering, yet upon reading what is a complete introduction, the reader gets the impression that the approach proposed by the authors is substantially more general than most previous applications. This impression is substantiated by the methodology presented in the following, which views HSHMs occurrence as a binary event in space/time and identifies indicator geostatistics as the tool of choice to adopt. A Bernoulli distribution is chosen to model the binary random variable, and this is due to a specific property of the Bernoulli distribution, which somewhat detracts from the generality of the model. A positive aspect of the methodology presented is the option to adopt different choices for activation thresholds: concentration, reactivity, mass, flux, percentiles, extremes. Three different HSHM categories are identified, i.e., induced by a) static, b) static and dynamic indicators, and c) multiple dynamic indicators. The stochastic formalism allows comparing alternative HSHM via information or Bayesian information criteria. Several examples are described via specific case studies for the three different categories and for groundwater hydrology. A more detailed illustrative example to groundwater hydrology based on classical stochastic hydrology theory valid under ergodicity is then illustrated. Wider pdfs of the indicator are associated with larger log-conductivity variances, and a convincing explanation is presented in detail. The appendix presents a recapitulation and/or extension of classical results in stochastic hydrology.*

*The paper looks as a mature contribution almost ready for publication. Given the topic and the type of paper, I see little room for further improvement. Results are of interest to the readership of Hydrology and Earth System Sciences. The methods are adequate, the paper subdivision into sections sound, and the figures illustrative. I reviewed only the revised version, but I can tell there were substantial improvements and clarifications upon looking at the earlier version and to the extent of the modifications. On the reviewers side, the remarks were extensive and provided a noteworthy input to the paper quality.*

*There is a general question that deserves the author's response, i.e.:*
— *How crucial is the assumption of a Bernoulli distribution for the indicator variable? Could they develop a more general theory without it, maybe subject to other limitations?*

Response: We thank the reviewer for the high-level summary of our work and positive remarks of our manuscript.
Based on our definition of HSHMs, any given pair of space and time component $(\boldsymbol{\Omega}^*, t)$ is whether a HSHM or not. The critical conditions that trigger HSHMs, however, are not necessarily binary and can take various forms. Our proposed framework allows the users to define different critical conditions based on their own data availability and expert knowledge. And these critical conditions can be updated if additional data becomes available.

*Minor correction: Note that at line 425, 4.3 -> 4.2*

Response: We have made the necessary correction (L430).

Response to RC#2

*Summary: The authors' response document and revised manuscript together helped me to better understand the work and to improve the manuscript. I thank them for that. In reading through the paper again, I must say I enjoyed it a lot more than the first time though. That may be due, in part, to being familiar with it, but I think the authors' responses and edits helped a lot. While I don't fully agree with everything, I think that's okay, we don't need to agree on everything. For example, I think I differ (maybe philosophically) from the authors in that I still see their framework as fundamentally binary that doesn't treat HSHMs in a continuous perspective. I realize that it is probabilistic and probabilities are continuous, but they are probabilities of a binary state as opposed to a continuous state. I am not suggesting the authors make any changes to their*

*framework, I think it is valuable and will push the field forward, and I look forward to further discussion in the literature on how to keep improving our understanding and quantification of HSHMs. I have only a few minor suggestions provided below.*

Response: We appreciate the reviewer's comments. The binary nature of the indicator stems from assuming the Hot Spots to be either present or not and the Hot Moments to be triggered only if certain conditions are observed or certain thresholds exceeded. This is somewhat implicit in the definition of HSHMs according to the literature and we are aware that as all classification approaches it aims at identifying where and when major changes occurs. However, this does not mean that processes are oversimplified or not included. As shown in the examples discussed in the manuscript the physical processes are simulated in a continuous manner, only the classification of zones (i.e. Hot Spots) and times (i.e., Hot Moments) of high activity is based on a binary random value.

Detailed comments:
*Line 115: I think Eq. 1 is saying that a HSHM can be trigged be either high concentration or high reaction rate. I am wondering if it's necessary to have a co-dependence between concentration and reaction rate? That is, if concentration is high, but reaction rate is low, then there won't be a HSHM, yes? Is that correct, or do the authors see it differently? It feels like the condition might require high concentration and high reaction as opposed to needing only high concentration or high reaction rate? At minimum, it's worth adding a couple sentences summarizing the logic of the setup and what deviations might be possible. Or maybe this is an issue of not really clearly defining what the authors mean by a HSHM prior to arriving at Eq. 1. I re-scanned the material above Eq. 1 and I didn't see a super clear definition. They lead off the paper talking about rates, so that's where my mind is. If the authors define changes in concentration alone (Without any change in rate) as a HSHM, it would be good to define that, potentially around Eq. 1.*

Response: We agree with the reviewer that in the previous version of the manuscript we were not clear about this point. The approach is able to address all the situations you mentioned and the ones to be considered, depending on the case at hand. In other words, the definition of the HSHM is built around the type of parameters to be considered. A case with high concentration and low reaction rate is a HMHS if the interest is to identify areas creating major risk of exposure, and on the other hand an area having high reactivity is an HSHM irrespective of the concentration, when the interest is in the removal, or regeneration, capacity of the system. Also the case with co-dependence between concentration and reaction rate can be treated with the proposed approach. In the revised manuscript, we have improved the clarity regarding when to use concentration based indicators or reaction rate based indicators. (L108-L109, L118-L122).

*Line 192: "interested" should be "intersected"?*

Response: We have made the necessary correction (L197).

*Line 377: The term "obviating" is used here, but I think the authors mean "requiring" yes? Or I may be deeply misunderstanding what they are driving at (i.e., if uncertainty obviates (i.e., removes) the need for stochastic modeling, I am confused).*

Response: We agree with the reviewer that we meant 'requiring'. We have made the necessary correction (L382).

*Fig. 5: I think there may be an error in the figure legend for e = values for dashed lines?*

Response: We apologized for the mislabeled legend. e = 0.1, 0.5 and 1.0 for the black, red and green dashed lines, respectively. We have corrected it in the revised manuscript (L475).

*Line 510: I don't understand the second bullet. Could it be removed? Seems duplicative with other bullets?*

Response: We have removed the second bullet.

[revised manuscript text omitted]